# Watermarking Language Models with Error Correcting Codes

## Abstract

Recent progress in large language models enables the creation of realistic machine-generated content. Watermarking is a promising approach to distinguish machine-generated text from human text, embedding statistical signals in the output that are ideally undetectable to humans. We propose a watermarking framework that encodes such signals through an error correcting code. Our method, termed robust binary code (RBC) watermark, introduces no noticeable degradation in quality. We evaluate our watermark on base and instruction fine-tuned models and find that our watermark is robust to edits, deletions, and translations. We provide an information-theoretic perspective on watermarking, a powerful statistical test for detection and for generating $p$-values, and theoretical guarantees. Our empirical findings suggest our watermark is fast, powerful, and robust, comparing favorably to the state-of-the-art.

## 1 Introduction

As language model capabilities improve, there are corresponding potential harms such as the creation of misinformation (Zellers et al., 2019) and propaganda (Solaiman et al., 2019). To mitigate such harms, a first step is to detect and filter content. A popular approach to reliably detecting AI generated content is to add a *watermark* (see e.g., Kirchenbauer et al., 2023; Kuditipudi et al., 2023; Aaronson and Kirchner, 2022; Christ et al., 2023; Hu et al., 2024; Huo et al., 2024, etc), a hidden signal embedded in the output that biases generation towards specific patterns that are undetectable to humans.

We consider the detection setting from the *model-provider*'s perspective: the detection algorithm receives (user or machine-generated) text as input, but no further metadata such as prompts or generation parameters. We explore how to introduce statistical signals into the generation process to reliably classify text as watermarked. We aim to design an effective watermark with the following properties, inspired by, e.g., (Kuditipudi et al., 2023; Kirchenbauer et al., 2023; Piet et al., 2025). The watermark should *not decrease the quality* of the outputs of the language model. The watermark should have *high detection power*, be detectable with only a small number of tokens. The watermark should be *robust* to watermarking attacks and perturbations, such as edits and deletions of significant fractions of the text, translation attacks, paraphrasing, etc.

Prior watermarking schemes achieve many but not all of these qualities. For instance, Kirchenbauer et al. (2023) partition the vocabulary into green and red sets based on a hash function of the context, and increase the sampling probability of green-listed tokens. While this logit bias introduces a detectable watermark signal, it may also degrade generation quality. Kuditipudi et al. (2023) introduces *distortion-free* watermarks that do not modify the output distribution in expectation, but their detection method typically requires permutation tests, which could be computationally expensive. Taking a cryptographic perspective, Christ et al. (2023) introduces theoretically *undetectable* watermarks, but their methods do not have robustness empirically. Christ and Gunn (2024) introduce *pseudorandom codes* (PRC) for watermarking and discussed their theoretical properties. But their watermark signal depends on fixed token positions, which makes the method vulnerable to edits that alter token indices, such as deletions. We aim to achieve the three desired properties by leveraging error correcting codes. While Qu et al. (2024) also use error correction codes to develop robust watermarks, their focus of embedding a multi-bit string (such as a user ID) in the text is different from ours. For further discussion of related work, see Appendix A.

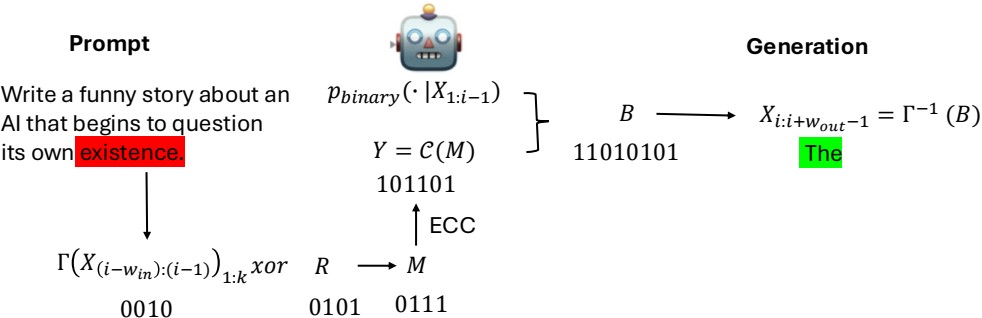

**Figure 1:** An overview of our method: at each generation step, the binary conversion of the previous tokens $\Gamma(X_{(i-w_{in}):(i-1)})$ is combined with a random bit string $R$ via an exclusive-or operation to construct the message $M$. The message is then encoded using an error-correcting code (ECC) to produce the codeword $Y$. The binarized language model generates binary strings through the Correlated Binary Sampling Channel (CBSC, Christ and Gunn, 2024) with codeword $Y$. Finally, the binary output is mapped back to the vocabulary space using the binary decoding function.

**Contributions.** Our paper introduces the **Robust Binary Code (RBC) watermark**, a novel watermark designed for reliable detection of machine-generated text. RBC leverages error-correcting codes (ECCs) combined with a sliding-window encoding mechanism to embed statistical signals effectively into generated content. Our contributions are as follows:

1. **Robustness via Error Correction.** RBC enhances watermark robustness by utilizing ECCs to encode messages derived from sliding-window token contexts. Empirically, RBC demonstrates superior resilience to substantial text perturbations, including random deletions, insertions, translations, and paraphrasing attacks, outperforming various baselines.
2. **Strong Detection Power.** We establish theoretical guarantees linking watermark detectability directly to language model entropy. Empirically, RBC achieves high detection power, even in short texts, outperforming prior distortion-free approaches, with detection probabilities higher across various text lengths.
3. **Minimal Distortion and High Quality.** Our RBC watermarking is next-token distortion-free, preserving the original distribution of the language model output at each generation step. Empirical evaluations show that RBC maintains generation quality in several metrics (e.g., perplexity) compared to watermarking methods that induce larger distribution shifts.

**Notation.** We use the following notation throughout this work. We typically use capital letters to denote random variables. For a positive integer $v$ and $p \in [0, 1]$, let $F_B(t; v, p)$ be the CDF of a Binomial($v, p$) random variable at the positive integer $t$, and let $[n] = \{1, \ldots, n\}$. Let $\mathcal{V}$ be the vocabulary of the language model, with $|\mathcal{V}|$ commonly between 50,000 to 200,000 in our current applications, and let $\mathcal{V}^*$ be the set of strings of tokens from $\mathcal{V}$ of arbitrary length. For a positive integer $i$, we use $X_i$ to denote the tokens generated by the language model $p : \mathcal{V}^* \to \Delta(\mathcal{V})$, and write $X_i \sim p(\cdot \mid x_{1:i-1})$ for any sequence $x_{1:i-1} \in \mathcal{V}^{i-1}$ of previous tokens. Let $p_j(\cdot \mid x_{1:i-1})$ be the distribution of the next $j$ tokens. For some positive integers $k, n, t$.

## 2  Background

### 2.1  Error Correcting Codes

To introduce our method, we need to provide some background on error correcting codes. For a positive integer $k$, we use the Hamming distance $d_H$ on $\{0, 1\}^k$, such that for $u, v \in \{0, 1\}^k$, $d_H(u, v)$ is the number of differing coordinates of $u, v$. We recall some standard definitions (MacWilliams and Sloane, 1977; Huffman and Pless, 2010).

**Definition 2.1.** *For positive integers $k \leq n$, an* error correcting *code (ECC) is an injective map $\mathcal{C} : \{0,1\}^k \to \{0,1\}^n$. The* message space *is $\{0,1\}^k$. Applying $\mathcal{C}$ to a message $m$ is known as* encoding, *and $\mathcal{C}(m)$ is known as a* codeword. *The* rate *of the code is $k/n$.*

*Let $\mathcal{R}(\mathcal{C})$ denote the range of the map $\mathcal{C}$. The* error correcting distance *of $\mathcal{C}$ is the greatest $t > 0$ such that for all binary vector $v \in \{0,1\}^n$, there exists at most one codeword $c \in \mathcal{R}(\mathcal{C})$ in the Hamming ball of radius of $t$ around $v$, i.e., $d_H(v, c) \leq t$. Such a code $\mathcal{C}$ is known as a $[n, k, 2t + 1]$-code.*

Given a $[n, k, 2t + 1]$-error correcting code $\mathcal{C}$, we may define a *decoding* map $\mathcal{C}^{-1}$. For a given $v \in \{0,1\}^n$, the decoding $\mathcal{C}^{-1}(v)$ is the preimage of the unique codeword $c \in \mathcal{R}(\mathcal{C})$ such that $d_H(v, c) \leq t$, if such a $c$ exists; otherwise, the decoding is defined arbitrarily. By definition, at most one such $c$ exists.

## 2.2 Reduction to Binary Vocabulary

Motivated by Christ et al. (2023), we reduce language modeling to a binary vocabulary. To do so, we assign to each token a unique bit string. For a vocabulary of size $|\mathcal{V}|$, this requires $\ell = \lceil \log_2 |\mathcal{V}| \rceil$ total bits; typically between 16 and 18 bits in our current applications. We use an injective binary converter $\Gamma : \mathcal{V} \to \{0,1\}^\ell$ that maps tokens to bit strings. A single token sampled from the language model $p(\cdot|x_{1:(a-1)})$ induces a distribution over the next $\ell$ bits, determined by the converter $\Gamma$. We use $q_{n+1} := p_{\text{bin}}(B_{n+1} = 1 \mid b_{1:n})$ to denote the probability of the next bit being one ("1") given previous bits $b_{1:n}$.

As in prior work (e.g., Aaronson and Kirchner, 2022; Kirchenbauer et al., 2023, etc), at each token generation, we leverage the previous tokens to change the distribution adaptively. Specifically, we use the context to construct a message $M \in \{0,1\}^k$, and use the error correcting code to obtain the codeword $Y = \mathcal{C}(M) \in \{0,1\}^n$. Then, we use the binary sampling scheme to generate bits $(B_1, \ldots, B_n)$ that are correlated with the $Y_i$s (as explained in Section 3). Each application of our watermarking scheme generates a block of $w_{\text{out}} > 0$ tokens, using the previous $w_{\text{in}} > 0$ tokens. This is related to prior approaches in neuro-linguistic steganography (Ziegler et al., 2019; de Witt et al., 2023), which use minimum-entropy couplings for correlated sampling between messages and covertext (de Witt et al., 2023). A brief overview of our method is provided in Figure 1.

## 2.3 Sampling Correlated Bits

Given a message $M \in \{0,1\}^k$ that we wish to transmit, we generate a codeword $Y = \mathcal{C}(M) \in \{0,1\}^n$. We now explain how to embed this codeword to sample bits $B \in \{0,1\}^n$ without inducing distortion. For this, we leverage the binary generation scheme from Christ and Gunn (2024). For each bit $B_i$, we need to sample a Bernoulli random variable. Suppose $B_i \sim \text{Bern}(q)$, $q \in [0,1]$ One can do this by first sampling a random variable $U' \sim \text{Unif}[0,1]$ and letting $B_i = 1\{U' \leq q\}$. Notably, $U'$ and $B_i$ are correlated.

An equivalent sampling scheme is to let $U' = (1 - Y_i)/2 + U/2$, where $Y_i \sim \text{Bern}(1/2)$ and $U \sim \text{Unif}[0,1]$ are sampled independently; so that $U' \sim \text{Unif}[0,1]$. By writing $U'$ in binary, $1 - Y_i$ represents the most significant bit of $U'$, and $U$ represents the remaining bits. Therefore, we have the following[1] sampling scheme for $B_i$:

$$Y_i \sim \text{Bern}(1/2), \ U \sim \text{Unif}[0,1], \ Y_i \perp\!\!\!\perp U, \quad B_i = 1\big\{\big((1 - Y_i) + U\big)/2 \leq q\big\}. \tag{1}$$

We call this sampling process the *Correlated Binary Sampling Channel* (CBSC), where we use an auxiliary random variable $Y_i \sim \text{Bern}(1/2)$ to sample a non-uniform bit $B_i \sim \text{Bern}(q)$, ensuring they are correlated.

**Next-token distortion-freeness.** Crucially, the CBSC ensures that $B_i$ matches its target distribution despite using the external information $Y_i$, i.e., $B_i \sim \text{Bern}(q)$. Thus, when applied at each generation step, this approach does not distort the distribution of each token given the previous block.

The matching probability between $Y_i$ and $B_i$ equals $\mathbb{P}[Y_i = B_i] = 1 - |1/2 - q|$ and is larger when the entropy of $B_i$ is larger. However, when $q \in \{0,1\}$—so $B_i$ has zero entropy—then $\mathbb{P}[Y_i = B_i] = 1/2$ and $Y_i$ contains no information about $B_i$. In general, the sampled bit $B_i$ is *biased towards $Y_i$* and is more likely to be equal to $Y_i$ than not.

---

[1]We use $1 - Y_i$ rather than simply $Y_i$ in determining $B_i$ so that $B_i$ and $Y_i$ are *positively correlated*; but the methods are equivalent.

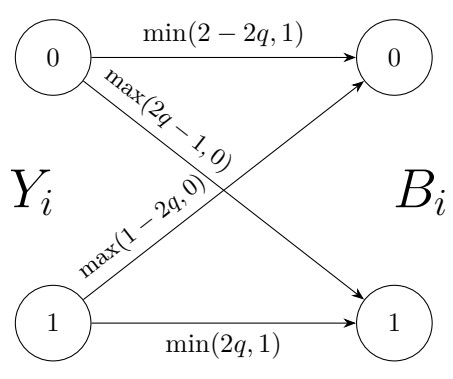
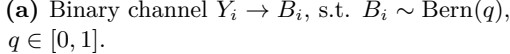
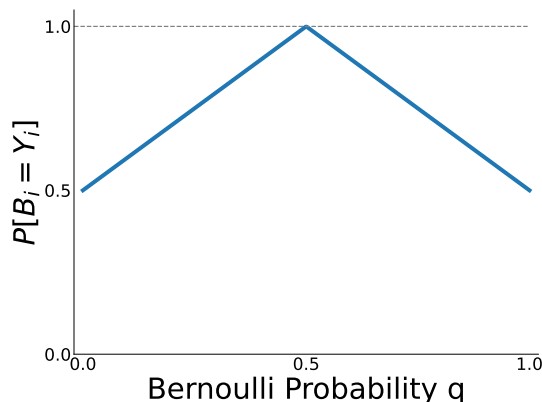

**(a)** Binary channel $Y_i \rightarrow B_i$, s.t. $B_i \sim \text{Bern}(q)$, $q \in [0, 1]$.

**(b)** Probability that $B_i = Y_i$, equal to $1 - |1/2 - q|$.

**Figure 2:** Correlated binary sampling channel (CBSC) with $Y_i \sim \text{Bern}(1/2)$ and $B_i \sim \text{Bern}(q)$.

## 3 Robust Binary Code Watermark

### 3.1 Simple Watermark

To develop our watermarking method, we will embed a sequence of *known* bits $Y = (Y_1, \ldots, Y_n)$ into our generated bits $B = (B_1, \ldots, B_n)$. Generating bits $B$ through the CBSC ensures $B$ follows the original probability distribution—i.e., is next-token distortion-free—and yet biases $B$ towards $Y$.

We may use this dependence to detect whether or not the text was watermarked. For unwatermarked text, $B$ and $Y$ are uncorrelated and therefore $d_H(B, Y) \approx n/2$, whereas for watermarked text, we expect $d_H(B, Y) < n/2$. This is the basis of our simple *one-to-one watermarking scheme*, described in Appendix B.2. We will next improve this by reducing errors in our binary channel via error correcting codes.

**Choosing known bits $Y$.** To reliably detect the watermark, we need to know the bits $Y$ encoded. While some works use one or several *secret sequences* known to the language model provider (see e.g., Kuditipudi et al., 2023, etc), this approach is not robust to deletion attacks. Instead, following Aaronson and Kirchner (2022); Kirchenbauer et al. (2023); Christ et al. (2023), we use a sliding window of previously generated tokens to serve as sources of randomness for $Y$, which amounts to leveraging *text-dependent* randomness (Piet et al., 2025) to generate pseudorandom bits $Y$.

**Error correcting codes in generation.** The purpose of using error correcting codes (ECCs) is to reduce mismatches between $B$ and $Y$. Rather than directly transmitting $Y = (Y_1, \ldots, Y_n)$, we add *redundancy* through a $[n, k, 2t + 1]$ ECC $\mathcal{C}$. We choose a shorter message $M \in \{0, 1\}^k$ and use $Y = \mathcal{C}(M) \in \{0, 1\}^n$. When using an ECC, we choose $M$ by hashing the previous bits, and then apply the error correcting code to $M$ to obtain our codeword $Y$. By using $\mathcal{C}$, we can correct $t$ errors in $Y$.

Combining these parts, our watermarking scheme contains the following steps, repeated until the stopping condition is satisfied:

1. **Create message:** Use previous tokens to create message $M \in \{0, 1\}^k$.
2. **Encode using $\mathcal{C}$:** Encode $M$ using ECC to obtain $Y = \mathcal{C}(M) \in \{0, 1\}^n$.
3. **Generate:** Transmit $Y$ using the correlated binary channel with probabilities from the language model to generate $B \in \{0, 1\}^n$.

We call each sequence of $n$ bits generated at an iteration a *block*. In Alg. 1 and Alg. 2, we present a simplified watermark and detection algorithm that captures the essence of our approach.

---

**Algorithm 1:** Simplified RBC Watermark

---

**Input:** Generation length $N$, binary language model $p_{\text{bin}}$, ECC $\mathcal{C}: \{0,1\}^k \to \{0,1\}^n$

**1** $X_{1:k} \sim p_{\text{bin},k}(\cdot)$ `// Initialize first k bits`
**2** $i \leftarrow k$
**3** **while** $i \leq N$ **do**
**4** $\quad M \leftarrow X_{(i-k+1):i} \in \{0,1\}^k$ `// Message`
**5** $\quad Y \leftarrow \mathcal{C}(M) \in \{0,1\}^n$ `// Codeword`
**6** $\quad$ **for** $j = 1$ to $n$ **do**
**7** $\quad\quad U_j \sim \text{Unif}[0,1]$ `// CBSC`
**8** $\quad\quad X_{i+j} \leftarrow \mathbb{1}\{(1 - Y_j + U_j)/2 \leq p_{\text{bin}}(\cdot \mid X_{1:i+j-1})\}$
**9** $\quad i \leftarrow i + n$
**10** **return** $X_{1:N}$;

---

**Algorithm 2:** Simplified Detection

---

**Input:** Bits $X_{1:N}$, threshold $T$

**1** **for** $i = 1$ to $N - n - k + 1$ **do**
**2** $\quad M \leftarrow X_{i:(i+k-1)}$
**3** $\quad B \leftarrow X_{(i+k):(i+n+k-1)}$
**4** $\quad \hat{M} \leftarrow \mathcal{C}^{-1}(B)$ `// Recovered message`
**5** $\quad Z_i \leftarrow k - d_H(\hat{M}, M)$ `// Matches`
**6** **if** $\sum Z_i > T$ **then**
**7** $\quad$ **return** Watermarked
**8** **else**
**9** $\quad$ **return** Not Watermarked

---

### 3.2 Full Watermark

Next, we present the full version of our watermark, which includes converting tokens to binary and back. Let $\ell = \lceil \log_2 |\mathcal{V}| \rceil$ be the number of bits for each token and let $w_{\text{in}} = \lceil k/\ell \rceil$ and $w_{\text{out}} = \lceil n/\ell \rceil$. Assume we have an injective *binary converter* $\Gamma : \mathcal{V} \to \{0,1\}^\ell$ which maps from tokens to bit strings, and the corresponding binary unconverter $\Gamma^{-1} : \{0,1\}^\ell \to \mathcal{V}$ (defined arbitrarily for strings outside of the image of $\Gamma$). We also have the language model $p_{\text{bin}}$ induced by $p$ over a binary alphabet, so that $p_{\text{bin}}(s_1, \ldots, s_n) := p_{\text{bin}}(S_{n+1} = 1 | s_1, \ldots, s_n)$ for any binary string $s_1, \ldots, s_n$. We let the converter operate on a list of tokens elementwise, and the unconverter operate on blocks of bits of length $\ell$.

Our full watermarking scheme is shown in Alg. 3, 4, 5, 6. Alg. 3, 4 expand on the simplified method from Alg. 1, and Alg. 5, 6 expand on the detection in Alg. 2. The RBC watermark adds the binary converter/unconverter $\Gamma$ to the simplified watermark in Section 3.1. In the generation of the messages $M$, we also apply an *exclusive or* (xor) between the previous bits with a randomly chosen bitstring $R \in \{0,1\}^k$ to ensure the message $M$ contains i.i.d. Bern(1/2) entries. In this step, we could also apply the minimum hashing method from Kirchenbauer et al. (2023).

---

**Algorithm 3:** RBC Watermarking

**Input:** Total token generation length $N$, window widths $w_{\text{in}}$ and $w_{\text{out}}$, unif. random binary string $R \in \{0,1\}^k$

1   $X_1, \ldots, X_{w_{\text{in}}} \sim p_{w_{\text{in}}}(\cdot)$
2   $i \leftarrow w_{\text{in}} + 1$
3   **while** $i \leq N$ **do**
4     $M \leftarrow \Gamma(X_{(i-w_{\text{in}}):(i-1)})_{1:k} \oplus r \in \{0,1\}^k$
5     $Y \leftarrow \mathcal{C}(M) \in \{0,1\}^n$
6     $X_{i:(i+w_{\text{out}}-1)} \leftarrow \texttt{GenBlock}(X_{1:(i-1)}, Y)$
7     $i \leftarrow i + w_{\text{out}}$
8   **return** $X$

---

**Algorithm 4:** GenBlock

**Input:** Previous text $X \in \mathbf{V}^*$, Codeword $Y \in \{0,1\}^n$, output tokens $w_{\text{out}}$

1   **for** $j = 1$ to $n$ **do**
2     $U_j \sim \text{Unif}[0,1]$
3     $B_j \leftarrow \mathbb{1}\{(1 - Y_j + U_j)/2 \leq p_{\text{bin}}(X, B_{1:(j-1)})\}$
4   **for** $j = n + 1$ to $w_{\text{out}} \cdot \ell$ **do**
5     $U_j \sim \text{Unif}[0,1]$
6     $B_j \leftarrow \mathbb{1}\{U_j \leq p_{\text{bin}}(X, B_{1:(j-1)})\}$
7   $B \leftarrow (B_1, \ldots, B_{w_{\text{out}} \cdot \ell})$
8   **return** $\Gamma^{-1}(B) \in \mathbf{V}^{w_{\text{out}}}$

---

**Algorithm 5:** Detection

**Input:** Tokens $X \in \mathcal{V}^*$, window widths $w_{\text{in}}$ and $w_{\text{out}}$, $\alpha$ level, binary string $R \in \{0,1\}^k$

1   $N \leftarrow$ length of $X$
2   **for** $i = 1$ to $N - w_{\text{out}} - w_{\text{in}} + 1$ **do**
3     $M_i \leftarrow \Gamma(X_{i:(i+w_{\text{in}}-1)}) \oplus R$
4     $B \leftarrow \Gamma(X_{(i+w_{\text{in}}):(i+w_{\text{in}}+w_{\text{out}}-1)})$
5     $\hat{M}_i \leftarrow \mathcal{C}^{-1}(B)$ // Extract Message
6     $Z_i \leftarrow k - d_H(\hat{M}_i, M_i)$ // Count matches
7   **return** $\text{BC}(Z_1, \ldots, Z_{N-w_{\text{out}}-w_{\text{in}}+1}, k)$

---

**Algorithm 6:** Binomial Comparison (BC) Test

**Input:** Matches $Z_1, \ldots, Z_m$, $k$ binomial size, $\alpha$ level

1   $p \leftarrow 1 - F_B(\sum_{i=1}^m Z_i, m \cdot k, \frac{1}{2})$
2   **if** $p \leq \alpha$ **then**
3     **return** WATERMARKED
4   **else**
5     **return** NOT WATERMARKED

---

## 4 Theoretical Results

We now provide theoretical guarantees for our methods, focusing on how the entropy of the language model affects watermarking. For the following results, we consider a language model with a binary vocabulary that generates bits $B_1, \ldots, B_n$. Let $q_i = \mathbb{P}[B_i = 1 | b_{1:i-1}] = p_{\text{bin}}(1 | b_{1:i-1})$ for any positive integer $i$ and bit string $b_{1:i-1}$. We provide the proofs of all of the following claims in Appendix D. We start with some preliminaries.

**Definition 4.1** (Entropy). *Let the entropy of a Bernoulli random variable $Y$ with success probability $q \in [0,1]$ be defined as $H(Y) := H(q) = -q \log_2 q - (1-q) \log_2(1-q)$, with $H(0) = H(1) = 0$. For a sequence of*

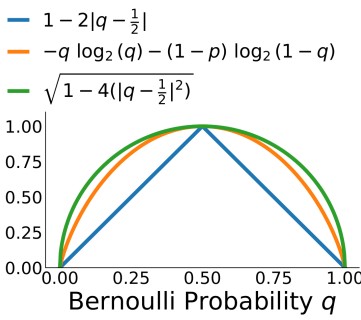

**Figure 3:** Bounds on $H(q)$ with functions of $|1/2 - q|$, for $q \in [0, 1]$.

(possibly dependent) Bernoulli random variables $Y_1, \ldots, Y_n$ with respective success probabilities $q_1, \ldots, q_n$, let the average entropy be $\bar{H}(Y_1, \ldots, Y_n) := \bar{H}(q_1, \ldots, q_n) := 1/n \cdot \sum_{i=1}^{n} H(Y_i)$.

We will need the following lemma, which bounds the entropy $H(q)$ in terms of $|1/2 - q|$, $q \in [0, 1]$.

**Lemma 4.2.** *For $q \in [0, 1]$, we have the inequality*

$$1 - 2|1/2 - q| \leq H(q) \leq \sqrt{1 - 4|1/2 - q|^2}.$$

*For $q_1, \ldots, q_n \in [0, 1]$, we have the inequality*

$$\frac{1}{n} \sum_{i=1}^{n} [1 - 2|1/2 - q_i|] \leq \bar{H}(q_1, \ldots, q_n) \leq \sqrt{1 - 4\left(\sum_{i=1}^{n} |1/2 - q_i|/n\right)^2}.$$

We plot the bounds on the entropy for the single-variate case in Figure 3. Next, using the above result, we turn to analyzing the CBSC. With $q_i = \mathbb{P}\left[B_i = 1 \mid b_{1:i-1}\right]$ for all $i$ and $b_{1:i-1}$, let the average entropy of the language model be $h := \mathbb{E}_{B,Y}\left[\frac{1}{n} \sum_{i=1}^{n} H(Q_i)\right]$. Intuitively, we expect that watermarking should be easier when the entropy $h$ is large ($h \approx 1$), as there are more ways to embed signals and keep the distribution unchanged. The next result supports this intuition.

**Theorem 4.3** (Bounding the proportion of mismatches in a CBSC with the entropy)**.** *The expected proportion of mismatches between the generated $B$ and codeword $Y$ —i.e., $\mathbb{E}\left[d_H(B, Y)/n\right]$ —is bounded by*

$$(1 - h)/2 \leq \mathbb{E}\left[d_H(B, Y)/n\right] \leq \sqrt{1 - h^2}/2.$$

*Moreover, we also have $\mathbb{E}\left[d_H(B, Y)/n\right] = n^{-1} \sum_{i=1}^{n} \mathbb{E}_{B,Y}\left[|1/2 - Q_i|\right]$.*

This theorem bounds the error probability via the average entropy $h = \mathbb{E}_{B,Y}\left[\sum_{i=1}^{n} H(Q_i)\right]/n$ of the language model, showing that the frequency of errors is both upper and lower bounded by a monotone decreasing function of $h$. In particular $\mathbb{E}\left[d_H(B, Y)/n\right] \to 0$ as $h \to 1$. This is consistent with watermarking being easier when the entropy is high.

The above result shows that the mean of $\Delta := \sum_{i=1}^{n} |1/2 - Q_i|$ is upper bounded by $n\sqrt{1 - h^2}/2$. Therefore, it is reasonable to assume that for some $C > 1$ and a small $\varepsilon_n > 0$, $\Delta \leq \kappa n \sqrt{1 - h^2}/2$ with probability at least $1 - \varepsilon_n$. Our next result will be presented under this condition.

**Theorem 4.4** (Exact block decoding)**.** *Consider a language model such that $q_i = \mathbb{P}\left[1 \mid B_{1:(i-1)}, Y\right]$ for $i \in [n]$. Let the average entropy of the language model be $h = \mathbb{E}_{B,Y}\left[\frac{1}{n} \sum_{i=1}^{n} H(Q_i)\right]$. Suppose that for some $C > 1$, $\Delta \leq \kappa n \sqrt{1 - h^2}/2$ with probability at least $1 - \varepsilon_n \in (0, 1)$. Assume $\mathcal{C} = [n, k, 2t + 1]$ is an error correcting code where $t + 1 \geq \kappa n \sqrt{1 - h^2}/2$. Then for a codeword $Y$, the generated bits $(B_1, \ldots, B_n)$ from Alg. 4 satisfy*

$$\mathbb{P}\left[\mathcal{C}^{-1}(B_1, \ldots, B_n) = \mathcal{C}^{-1}(Y)\right] \geq 1 - \exp\left[-\left(t + 1 - \kappa n \sqrt{1 - h^2}/2\right)^2/n\right] - \varepsilon_n.$$

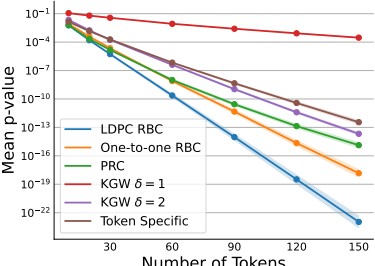 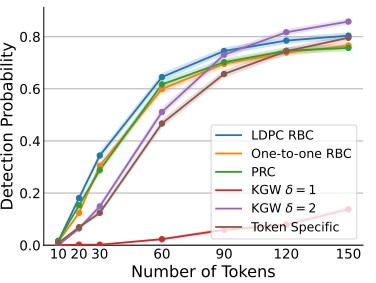 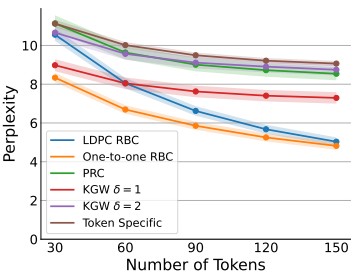

**Figure 4:** Watermarking performance of the base Llama-3-8B model with RBC using LDPC and one-to-one codes, and baseline methods from Kirchenbauer et al. (2023); Huo et al. (2024), averaged over 100 generations for ten prompts. **Left:** The mean log-$p$-values with standard errors shaded. **Middle:** The detection probability with standard errors shaded. **Right:** The mean perplexity of the generated texts with standard errors shaded.

The above theorem bounds the probability that a specific block *exactly* decodes to the codeword $Y$. In particular, it gives a bound where the decoding block probability decreases as the entropy of the language model decreases. In particular if $t/n \gg \sqrt{1 - h^2}$ and $\varepsilon_n$ is small, decoding is correct with high probability. This is consistent with the fact that watermarking is more difficult for models with less entropy when $h$ is small, in which case $t/n$ needs to be large.

## 5 Experimental Results

We evaluate the performance of RBC in terms of detection power, robustness to perturbations, and generation quality. We compare it against both distortion-free and non-distortion-free methods.

**Generation procedure.** We use ten prompts inspired by Piet et al. (2025), encompassing topics such as writing book reports, stories, and fake news articles. Piet et al. (2025) used these text generation tasks to represent realistic settings with long-form generated text, where the model provider may like to watermark the outputs. We provide the full prompts used for generation in Appendix E.

We generate 100 responses for each prompt for seven lengths ranging from 10 to 150 tokens, representing a few sentences to a few paragraphs. We use the Llama-3-8B base model (AI@Meta, 2024), a powerful open-source model capable of high quality generation. Additional experiments using the OPT-1.3B (Zhang et al., 2022), Mistral-7B models (Jiang et al., 2023) and the Llama-3-8B-Instruct model are reported in Appendix C.2 and Appendix C.5.

**Hyperparameter setting for our method**. There are many ECCs, with varying guarantees (Bose and Ray-Chaudhuri, 1960; Gallager, 1962; Reed and Solomon, 1960). Different ECCs provide different guarantees, e.g., robustness to bit flips or edit distance. For this work, we explore two ECCs. We primarily use the popular low-density parity-check (LDPC) code (Gallager, 1962) for our watermarking scheme. As a baseline, we also use a custom "one-to-one code", which has $\mathcal{C}(x) = x$ for all $x$; this does not provide error correcting capabilities (See Appendix B.2 for more details). We use this as a baseline to compare the impact of the error correcting functionality. We use the LDPC code (Shokrollahi, 2004) with $n = 12$, $k = 5$, $d_v = 3$, $d_c = 4$, and a one-to-one code with $n = k = 4$, using $w_{\text{in}} = 1$ and $w_{\text{out}} = 1$. For the LDPC code, we use the Belief Propagation algorithm for the Binary Symmetric Channel in decoding.[2] We set the probability of bit error for the Binary Symmetric Channel used in decoding to be $p = 0.35$. We set $\Gamma$ to be a random binary enumeration. Our method is fairly robust to hyperparameter choices. Additional results using alternative hyper-parameter settings are presented in Appendix C.4.

---

[2]The decoding method is not exactly $\mathcal{C}^{-1}$. We found that this decoding method works well in practice. The optimal detection algorithm would be to find the MAP (maximum a posteriori) estimator under the Correlated Binary Sampling Channel (CBSC), which would require access to the bit probabilities of the binarized language model. This optimal detection method requires evaluating/accessing the language model in the detection stage with known prompts, which will be computationally expensive and not always practical (e.g., the prompts may not be available in practice for detection).

**Baselines and hyperparameter setting.** We compare the performance of RBC to distortion-free watermarking methods. We consider the method from Kuditipudi et al. (2023), which employs exponential minimum sampling (Aaronson and Kirchner, 2022), and incorporates a soft notion of edit distance (Levenshtein distance) into the computation of the test statistic to improve robustness. We set the number of keys to 256 and use permutation tests with 100 resamples. We refer to this method as *EXP-Edit*. We also compare our method with the Pseudorandom Error-Correcting Codes (Christ and Gunn, 2024, PRC). This approach is similar to ours but replaces the sliding window mechanism with the generation of a long sequence of codewords based on a secret key. We use the LDPC code with the same settings as our method for PRC.

We also consider the non-distortion-free baseline method from Kirchenbauer et al. (2023), which we denote by *KGW*. The thorough evaluation from Piet et al. (2025) finds this to be the most practically effective and powerful watermark, and concludes that it serves as the state-of-the-art. We also compare with the method from Huo et al. (2024), a variant of the method Kirchenbauer et al. (2023), which we denote as *Token Specific*.

For the methods from Kirchenbauer et al. (2023) and Huo et al. (2024), we use their suggested hyperparameters. A key parameter is $\delta$, the level of distribution shift induced by the watermarking procedure. Greater values of $\delta$ result in larger distribution shifts and more detectable watermarking. The authors of Kirchenbauer et al. (2023) recommend $\delta \in [0.5, 2]$ for watermarking. In our experiments, we choose to use $\delta = 1$ and 2 to represent medium and strong distribution shifts for the method from Kirchenbauer et al. (2023). For the method from Huo et al. (2024), a lightweight neural network is trained to predict token-specific $\delta$ and $\gamma$ values, where $\gamma$ controls the fraction of tokens in the green list. We set the network parameters using the default checkpoint suggested by the authors[3].

**Detection Power.** For each generation, we evaluate the detection power using the $p$-value of the detection procedure applied to the output text. We compute the exponential of the mean of the log-$p$-values (instead of the $p$-values themselves) for each generation to more accurately capture the signal strength, as the $p$-values are mostly close to zero, and hence their means are not informative. We also report the detection probability, i.e., true positive rate or power, the percentage of generations with $p$-values less than $\alpha = 10^{-6}$, to represent how likely the text is to be classified as watermarked.[4] An empirical comparison between the $p$-value threshold and empirical false positive rate (FPR) can be found in Appendix C.1. Our results show that the empirical FPR closely aligns with the $p$-value thresholds.

From the summary statistics in Table 1 and Figure 4, we observe that our RBC watermarking method significantly outperforms the distortion-free baseline EXP-Edit in terms of detectability. It achieves detection power comparable to KGW with $\delta = 2$ on long sequences, and demonstrates improvements on shorter sequences. Further, the median and mean of $p$-values produced by our method are several orders of magnitude smaller than those from other approaches. Additionally, RBC improves performance by using the LDPC code compared to the one-to-one code, suggesting improvements by using an ECC.

**Robustness**. In practice, a user may attempt to circumvent watermarks by editing the text generated by a language model. To emulate this, we use four popular perturbations that may represent an adversary hoping to evade a watermark, as in other works including Kuditipudi et al. (2023); Piet et al. (2025).

1. **Delete.** We randomly delete 20% of the tokens.
2. **Swap.** We replace a randomly chosen fraction of 20% of the tokens with random tokens.
3. **Translate.** We translate the generated text to Russian then back to English using Argos Translate (Finlay and Argos Translate, 2023).
4. **Paraphrase.** We paraphrase the text using the same model as that used to generate text, with the prompt "`Paraphrase the following text:`".

Our goal is to evaluate robustness under model-agnostic transformations that approximate realistic misuse scenarios. Designing adaptive, watermark-aware attacks is a nontrivial problem and is typically treated as a separate line of work, which is beyond the scope of this paper.

---

[3]`https://github.com/mignonjia/TS_watermark/tree/main/ckpt/llama`
[4]For EXP-Edit, we use 100 resamples for the permutation test and a sequence of 256 secret keys. The $p$-value is at least 1/101. We set the detection threshold to $\alpha = 10^{-2}$, but the method still exhibits low detection power. Due to the high computational complexity of the permutation test and low detection power, we only include it in Table 1.

**Table 1:** Comparison between our watermarking methods and distortion free method for 30 and 150 tokens. We report the exponential of mean log $p$-value, the median $p$-value, and the percentage of generations with $p$-values less than $\alpha = 10^{-6}$. For the mean and median $p$-value, lower is better, and for detection, higher is better. The best value in each column is **bolded**.

| Method | 30 Tokens | | | 150 Tokens | | |
|---|---|---|---|---|---|---|
| | Mean $P$ | Median $P$ | Detect % | Mean $P$ | Median $P$ | Detect % |
| LDPC RBC | **5.5e−6** | **1.4e−5** | **34.4** | **1.1e−23** | **9.3e−20** | 80.4 |
| One-to-one RBC | 2.3e−5 | 6.0e−5 | 30.5 | 1.5e−18 | 1.9e−16 | 76.6 |
| EXP-Edit | 1.3e−1 | 2.0e−1 | 16.1 | 5.2e−2 | 5.0e−2 | 30.5 |
| PRC | 1.5e−5 | 2.9e−5 | 28.8 | 1.3e−15 | 2.2e−14 | 75.7 |
| KGW $\delta = 1$ | 3.7e−2 | 7.4e−2 | 0.2 | 3.0e−4 | 6.6e−4 | 13.8 |
| KGW $\delta = 2$ | 1.7e−4 | 6.4e−4 | 14.9 | 2.1e−14 | 3.8e−14 | **85.8** |
| Token Specific | 1.9e−4 | 4.5e−4 | 12.4 | 3.5e−13 | 2.6e−12 | 79.6 |

**Table 2:** Comparison between our watermarking methods and the baselines with different perturbations.

| | Method | 30 Tokens | | | 150 Tokens | | |
|---|---|---|---|---|---|---|---|
| | | Mean $P$ | Median $P$ | Detect % | Mean $P$ | Median $P$ | Detect % |
| *Swap* | LDPC RBC | 2.6e−3 | 6.2e−3 | 4.6 | **5.3e−10** | **1.0e−8** | **59.0** |
| | One-to-one RBC | 3.8e−3 | 9.9e−3 | 4.7 | 4.6e−8 | 5.8e−7 | 51.3 |
| | PRC | **8.8e−4** | **2.3e−3** | **9.9** | 2.4e−7 | 3.4e−5 | 41.5 |
| | KGW $\delta = 1$ | 1.1e−1 | 1.6e−1 | 0.0 | 9.8e−3 | 2.1e−2 | 1.7 |
| | KGW $\delta = 2$ | 8.0e−3 | 2.1e−2 | 1.7 | 4.1e−7 | 1.0e−6 | 49.8 |
| | Token Specific | 9.2e−3 | 2.4e−2 | 1.5 | 4.2e−6 | 1.5e−5 | 35.1 |
| *Delete* | LDPC RBC | **7.8e−4** | **2.2e−3** | 9.0 | **8.2e−13** | **2.5e−10** | **67.9** |
| | One-to-one RBC | 1.9e−3 | 4.4e−3 | 6.4 | 3.5e−10 | 8.7e−9 | 60.4 |
| | PRC | 2.0e−1 | 3.5e−1 | 0.2 | 2.9e−1 | 4.2e−1 | 0.0 |
| | KGW $\delta = 1$ | 8.8e−2 | 1.5e−1 | 0.2 | 4.9e−3 | 1.2e−2 | 4.6 |
| | KGW $\delta = 2$ | 4.2e−3 | 9.2e−3 | 3.1 | 1.5e−8 | 4.1e−8 | 64.2 |
| | Token Specific | 4.3e−3 | 9.0e−3 | 3.0 | 1.5e−7 | 7.6e−7 | 51.1 |
| *Translate* | LDPC RBC | **5.3e−3** | **1.9e−2** | **3.4** | **6.1e−10** | **1.3e−6** | **48.6** |
| | One-to-one RBC | 9.1e−3 | 2.3e−2 | 2.8 | 4.5e−8 | 6.8e−6 | 44.5 |
| | PRC | 5.2e−2 | 1.2e−1 | 0.9 | 9.3e−2 | 2.2e−1 | 0.8 |
| | KGW $\delta = 1$ | 1.3e−1 | 2.1e−1 | 0.0 | 1.9e−2 | 4.0e−2 | 1.2 |
| | KGW $\delta = 2$ | 1.6e−2 | 3.0e−2 | 1.5 | 5.3e−6 | 1.3e−5 | 35.5 |
| | Token Specific | 2.2e−2 | 4.7e−2 | 0.6 | 4.2e−5 | 2.1e−4 | 23.1 |
| *Paraphrase* | LDPC RBC | **2.4e−3** | 1.1e−1 | **11.1** | **1.2e−7** | 3.5e−3 | **37.6** |
| | One-to-one RBC | 6.0e−3 | **1.0e−1** | 8.4 | 8.0e−7 | **1.4e−3** | 37.2 |
| | PRC | 1.0e−1 | 3.3e−1 | 1.2 | 1.0e−3 | 2.9e−1 | 15.8 |
| | KGW $\delta = 1$ | 2.1e−1 | 4.2e−1 | 0.0 | 4.1e−2 | 1.6e−1 | 1.0 |
| | KGW $\delta = 2$ | 5.2e−2 | 2.2e−1 | 2.2 | 3.7e−5 | 4.9e−3 | 31.9 |
| | Token Specific | 2.1e−2 | 2.4e−1 | 5.4 | 8.3e−5 | 8.6e−3 | 28.2 |

For our experiments, we first show the results when perturbing 20% of the tokens. Results with other rates of perturbation can be found in Appendix C.3. In our translation perturbation, we translate to Russian and then back to English. In our paraphrase perturbation, we also consider using the more powerful GPT-4o model to paraphrase the text. The results can be found in Appendix C.3.

In Table 2 and Figure 5, we evaluate the robustness of RBC and baseline watermarking schemes under various perturbations. Notably, both the LDPC and one-to-one RBC watermarks demonstrate greater robustness than the baseline methods. Under deletion perturbations, the baseline method PRC completely fails to detect the watermark. Intuitively, this is because once a token is deleted, all subsequent tokens are misaligned and compared against incorrect codewords during detection.

**Generation Quality.** As discussed in Section A.1, our method is next-token distortion-free, which provides theoretical guarantees on preserving generation quality. We further validate this by evaluating the perplexity

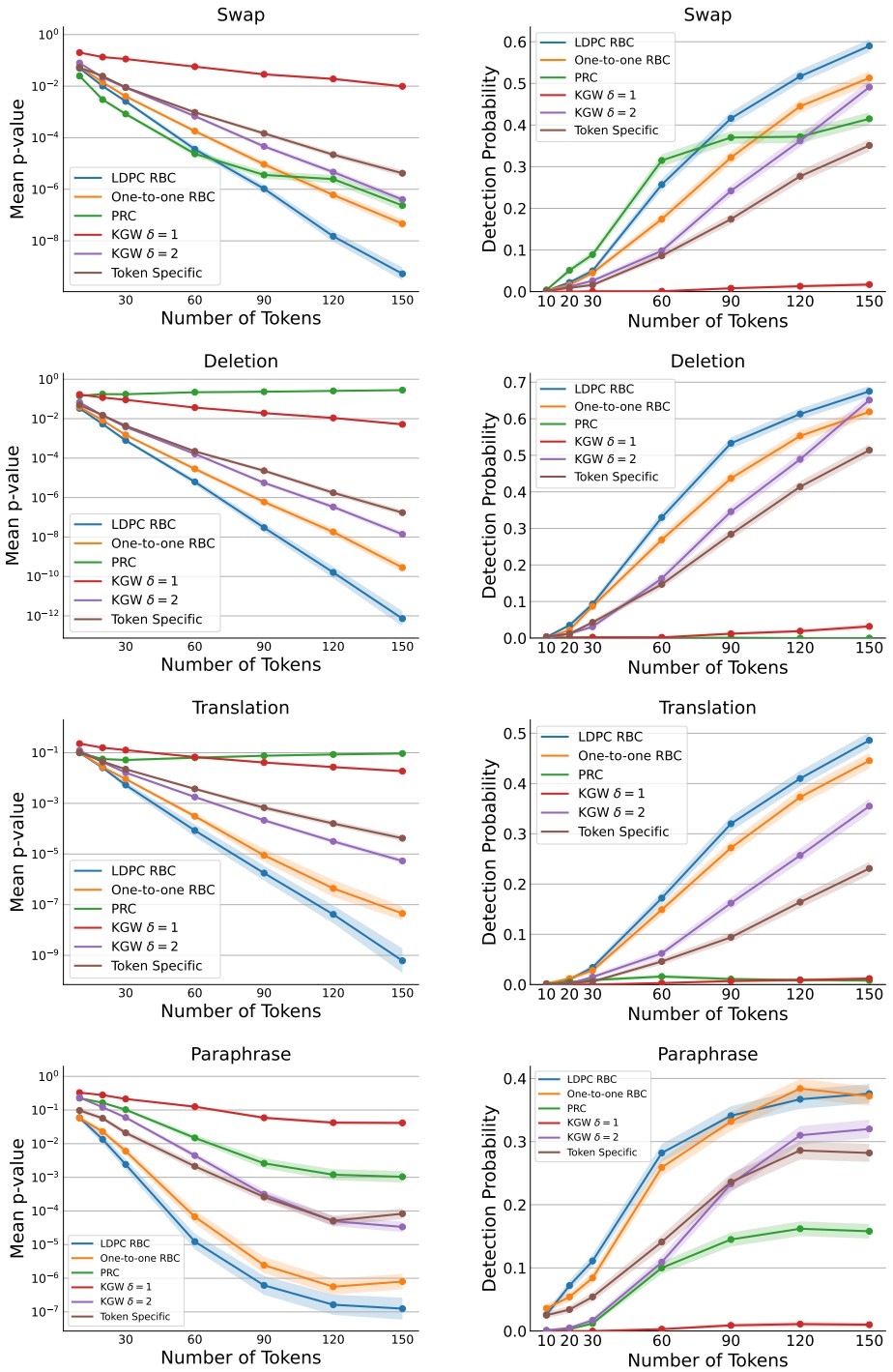

**Figure 5:** Watermarking performance of the base Llama-3-8B model with RBC using LDPC and one-to-one codes, and baseline methods. **Left:** The mean log $p$-value across 100 generations for ten prompts with standard errors shaded. **Right:** The detection probability with $\alpha = 10^{-6}$ with standard errors shaded. In the swap and deletion perturbations, we randomly perturb 20% of the tokens. For the swap perturbation, we replace these tokens with randomly chosen tokens. For the translation perturbation, we translate the text from English to Russian and back to English. For the paraphrase perturbation, we paraphrase the text using the same Llama-3-8B model.

of the generated text [5]. Perplexity is a commonly used metric to evaluate the generation quality for watermarking methods (Kirchenbauer et al., 2023), and more generally, of language models. Additional results using factual accuracy for instruction-fine-tuned models are provided in Appendix C.5.

Figure 4 (right) shows that our methods achieve good generation quality with low perplexities, especially for long sequences. Although KGW with $\delta = 2$ demonstrates strong detection power and robustness, the large distortion introduced by $\delta$ leads to low generation quality with higher perplexity.

**Summary.** Our watermarking method, RBC, consistently performs well across multiple critical metrics, including detection power, robustness to perturbations, and generation quality. In particular, RBC achieves significantly lower mean and median log-$p$-values, demonstrating substantially improved detection capabilities, especially notable in shorter text sequences compared to baseline methods such as KGW and EXP-Edit. Importantly, our method maintains high detection probability even after aggressive text perturbations, including random token swaps, deletions, translations, and paraphrasing, clearly establishing superior robustness. Moreover, RBC's use of LDPC error-correcting codes consistently enhances performance relative to simpler encoding schemes, highlighting the critical role of ECC in watermark effectiveness. Furthermore, our approach preserves text generation quality, as evidenced by lower perplexity scores, aligning with our theoretical guarantees on distortion-free next-token selection. Thus, RBC emerges as a superior watermarking solution, effectively balancing robust detection with high-quality, minimally-distorted text generation.

## 6 Discussion

We proposed a new watermark with high detection power and text quality, which is further robust to perturbations. While our preliminary experiments are promising, in future work it will be important to evaluate broader classes of perturbations, generation tasks, and language models. Additionally, it remains to evaluate the many possible choices of ECCs, e.g., BCH, Turbo codes, etc. Since our work aims to develop watermarking methods, it may have a positive social impact by enabling the reliable identification of LLM-generated text.

---

[5]In our experiment, we use the same model for generation and evaluation of the perplexity

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

# A  Further Related Work

There is a great deal of work on watermarking language models, see e.g., Yang et al. (2025) for an overview. Here we only review the most closely related works.

**Bias-based watermarking.**  In modern watermarking, one of the earliest and most widely adopted techniques involves biasing the model's token distribution to imprint a detectable signal. Aaronson and Kirchner (2022) proposed a method using exponential minimum sampling guided by pseudorandom hashes of previous tokens. Kirchenbauer et al. (2023) introduced a family of bias-based watermarks that divide the token vocabulary into green and red sets (using a hash function on the context), and increase the sampling probability of green-listed tokens. This logit bias adds a detectable signature without substantially degrading output quality. Qu et al. (2024) combined the watermarking methods in Kirchenbauer et al. (2023) with error correction codes to embed a multi-bit string (such as a user ID) in the text. In contrast, our focus is to detect if the text is watermarked or not, which corresponds to embedding a single-bit string. In the single-bit setting, the method from Qu et al. (2024) is equivalent to the method from Kirchenbauer et al. (2023).

Subsequent improvements have aimed to address detection power and robustness. Huang et al. (2024) developed SynthID-Text, a scalable production watermark that modifies token sampling with imperceptible biases. Zhao et al. (2023) proposed a distortion-robust watermark with edit-distance guarantees, using a fixed green-list across positions. However, as Jovanović et al. (2024) argued, fixed green-lists are vulnerable to reverse-engineering attacks.

**Distortion-free and cryptographic watermarks.**  To address the vulnerability of bias-based watermarks to detection or removal, Christ et al. (2023) introduced the theoretical notion of *undetectable* watermarks—text that is indistinguishable from natural output with polynomially many queries unless one holds the secret key. This approach is related to *steganography*, which has been of interest for many years (Katzenbeisser and Petitcolas, 1999; Atallah et al., 2001). Their approach leverages cryptographic pseudorandom functions to correlate sampled tokens with hidden keys while preserving the output distribution. They did not develop empirical algorithms for real language models.

Kuditipudi et al. (2023) proposed *robust distortion-free watermarks* that match the original LLM distribution in expectation (though not with a cryptographic level of strength) and provide robustness to small edits via testing. However, since this work uses permutation tests, it can be expensive to obtain small $p$-values required for high power. Xie et al. (2024) used maximal coupling to eliminate logit bias entirely while still embedding a watermark signal.

In the work Christ and Gunn (2024) (which was concurrent to the initial development of our research), the authors introduced *pseudorandom codes* (PRC) for watermarking, leveraging the same CBSC that we also independently developed in our work. In their algorithm development, they concatenate fixed-length bit strings generated from PRCs, and then use the CBSC to generate from the binarized language model using the bit string as the key. In contrast, our method used a sliding window approach to generate the key, which leads to enhanced robustness to deletions, as the watermark signal can be recovered even if certain tokens are deleted (which is not the case if one follows the approach from Christ and Gunn (2024) that uses fixed bit strings from PRCs). Moreover, while Christ and Gunn (2024) provides detailed theoretical discussion, it does not give empirical implementation details or evaluation results on language models. In contrast, our work provides detailed implementations (including the choice of both LDPC and one-to-one codes), hyperparameter choices, and experimental results both for detection power and robustness, comparing to a broad range of baselines.

**Robustness and attacks.**  Robustness to paraphrasing, insertion, or translation is critical in adversarial settings. Krishna et al. (2023) showed that paraphrasing substantially reduces watermark detection rates. Hou et al. (2023; 2024) proposed semantic watermarks that insert paraphrastic synonyms selected to survive rewording. Liu et al. (2024) developed a semantic-invariant watermark that maps paraphrases to the same green-list via learned encodings.

Despite these defenses, several works have shown vulnerabilities. Chang et al. (2024) proposed smoothing attacks using weaker models to paraphrase and erase watermarks. Sadasivan et al. (2024) introduced recursive paraphrasing to degrade detection iteratively. He et al. (2024) demonstrated that translation to another language and back often removes the watermark. Jovanović et al. (2024) highlighted that many watermarking methods can be reverse-engineered or spoofed by observing enough outputs.

### A.1  Discussion of distortion-freeness

We propose a generation technique to encode signals that is distortion-free given each previous block of tokens of a certain size. Of course, this technique is still not distortion-free over the entire generation. Our method relies on a pseudorandom number generator influenced by a finite window of past tokens, and as such, the entropy of the random numbers can degrade if the past tokens exhibit low entropy. Thus, our technique does not achieve perfect security from a steganographic perspective (Cachin, 1998; de Witt et al., 2023).

## B  Implementation Details

### B.1  Alternative Statistical Tests

We also explore alternative statistical tests to the simple binary comparison test in Alg. 6. These use the generalized likelihood ratio test, and use Wilks's theorem (Wilks, 1938) to approximate the distribution of the log-likelihood ratio by a $\chi^2$ distribution.

---

**Algorithm 7:** Generalized Likelihood Ratio Test (GLRT)

**Input:** Matches $Z_1, \ldots, Z_m$, number of trials in a binomial random variable $k$

1 $\boldsymbol{L}_0 \leftarrow \sum_{i=1}^m \ln f_B(Z_i; k, 1/2)$
2 $\boldsymbol{L}_1 \leftarrow \sum_{i=1}^m \ln f_B(Z_i; k, Z_i/k)$
3 $T \leftarrow -2(L_0 - L_1)$
4 $P \leftarrow 1 - F_{\chi^2}(T; \mathrm{df} = m)$
5 **return** $P$

---

**Algorithm 8:** Pooled Generalized Likelihood Ratio Test (PGLRT)

**Input:** Matches $Z_1, \ldots, Z_m$, number of trials in a binomial random variable $k$

1 $\hat{q} \leftarrow \sum_{i=1}^m Z_i/(k \cdot m)$ $\boldsymbol{L}_0 \leftarrow \sum_{i=1}^m \ln f_B(Z_i; k, 1/2)$
2 $\boldsymbol{L}_1 \leftarrow \sum_{i=1}^m \ln f_B(Z_i; k, \hat{q})$
3 $T \leftarrow -2(L_0 - L_1)$
4 $P \leftarrow 1 - F_{\chi^2}(T; \mathrm{df} = 1)$
5 **return** $P$

---

### B.2  One-to-one Code

Our one-to-one code $\mathcal{C} : \{0, 1\}^k \rightarrow \{0, 1\}^k$ maps between length $k$ bit strings, and is defined as $\mathcal{C}(m) = m \oplus R$, where $R$ is a fixed bit string. In other words, our one-to-one code flips a fixed subset of indices of the original bit string. We choose $R$ with i.i.d. Bern(1/2) entries, so that $\mathcal{C}(m)$ also contains random entries.

## C  Additional Experimental Results

### C.1  Empirical False Positive Rates and p-values

Our method uses the Binomial Comparison (BC) test to compute $p$-values and detect watermarks by comparing each $p$-value against a threshold $\alpha$. In this section, we evaluate how the $p$-value threshold corresponds to

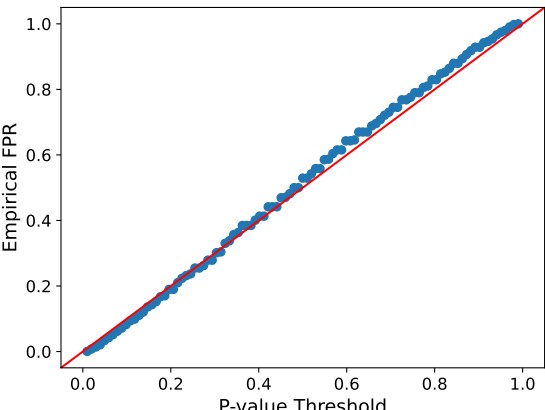

**Figure 6:** Empirical FPR vs P-value thresholds.

the empirical false positive rate (FPR, the proportion of unwatermarked text detected as watermarked). We consider the same setting as in Table 1. We use the Llama-3-8B model to generate 100 unwatermarked responses for each of 10 prompts and evaluate the FPR across different $p$-value thresholds. As shown in Figure 6, the empirical FPR closely aligns with the $p$-value thresholds. The results indicate that the $p$-value threshold accurately reflects the false positive rate.

### C.2   Results for different LLMs

Figure 7 and Figure 8 show the mean $p$-value and detection rate of different watermarking schemes using the OPT-1.3B (Zhang et al., 2022), Mistral-7B-v0.1 (Jiang et al., 2023) model. Our method achieves better detection power under various perturbations.

### C.3   Results with additional perturbations

For the Llama-3-8B model, we further evaluate robustness under additional perturbation attacks, including Swap and Delete operations at varying perturbation rates, as well as a paraphrasing attack using GPT-4o. The results are summarized in Table 3. Under strong perturbations—such as paraphrasing with the more powerful GPT-4o model or high-rate Swap and Delete attacks (e.g., 50%)—all watermarking methods exhibit reduced detection performance. Nevertheless, our method achieves the best performance across most experimental settings.

### C.4   Results with different hyper-parameters

We provide a detailed study of the effect of hyperparameters. Let $k, n$ be the number of bits in $M$ and $Y = \mathcal{C}\,(M)$, where $\mathcal{C}$ is the error correcting code. For the one-to-one code, we report results with various values of $n = k$. For the LDPC code, we consider the code with $d_v = 3, d_c = 4$. We present results across different values of $n$ (with $k$ determined by the code). We also evaluate alternative settings $d_v = 2, d_c = 4$ and $d_v = 3, d_c = 6$, and report results corresponding to the best-performing values of $n$. For LDPC decoding, we adopt the Belief Propagation algorithm under a Binary Symmetric Channel model with bit-error probability $p$, and report results across different choices of $p$. We also include a comparison with exact decoding [6] via $\mathcal{C}^{-1}$. Table 4 shows that our method is fairly robust to the hyper-parameter settings. The parameters listed in the "Method" column are the ones changed from the default setting $n = 12, d_v = 3, d_c = 4, p = 0.35$.

---

[6]Intuitively, the bit string $B$ used for decoding is different from $Y$. It is generated by transmitting $Y$ through the correlated binary channel with probabilities extracted from the language model. Therefore, applying exact decoding $\mathcal{C}^{-1}(B)$ to match with $M$ is not necessarily optimal.

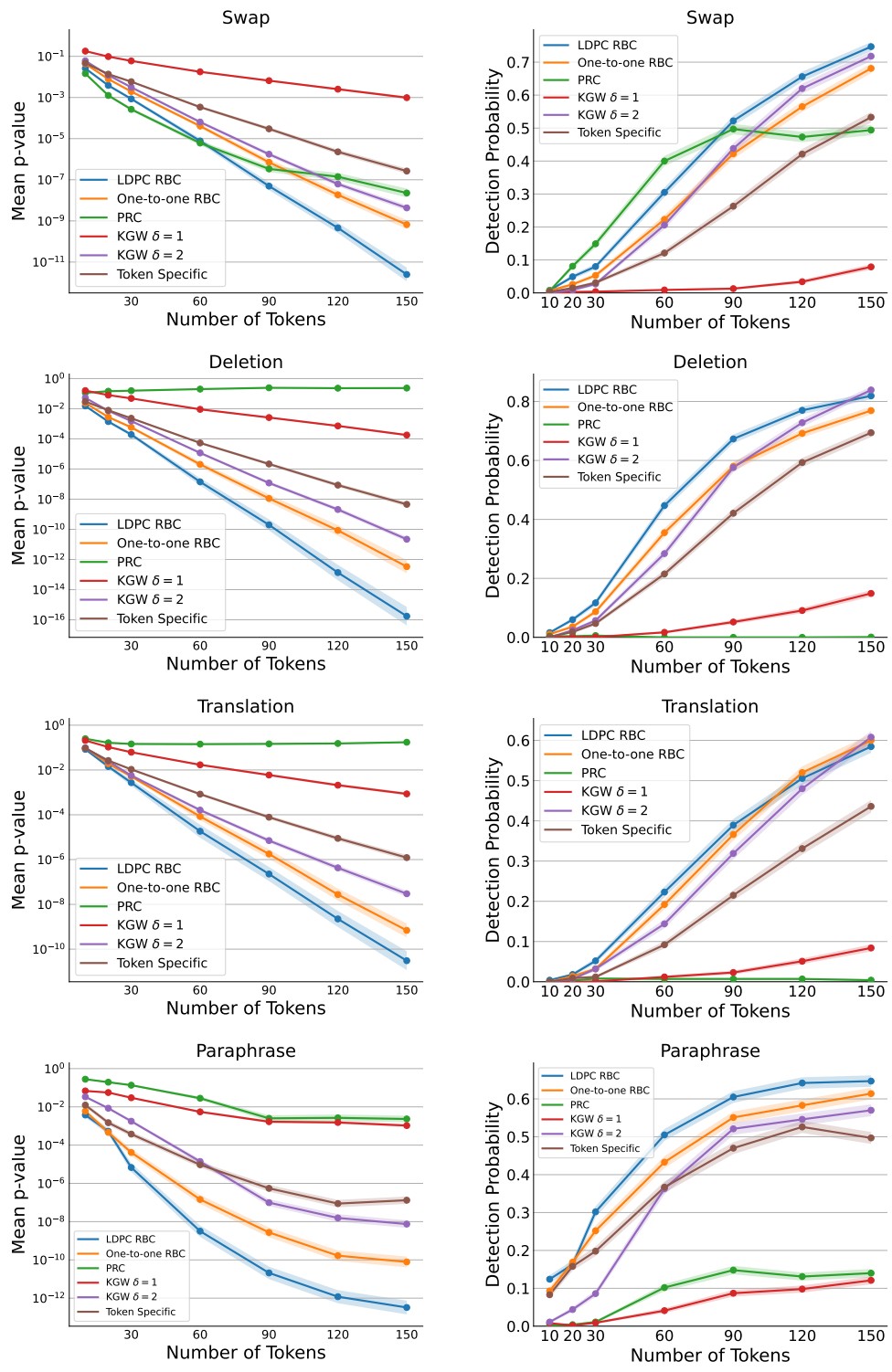

**Figure 7:** Watermarking performance for the OPT-1.3B model with RBC using LDPC and one-to-one codes, and baseline methods. **Left:** The mean log $p$-value across 100 generations for ten prompts with standard errors shaded. **Right:** The detection probability with $\alpha = 10^{-6}$, with standard errors shaded.

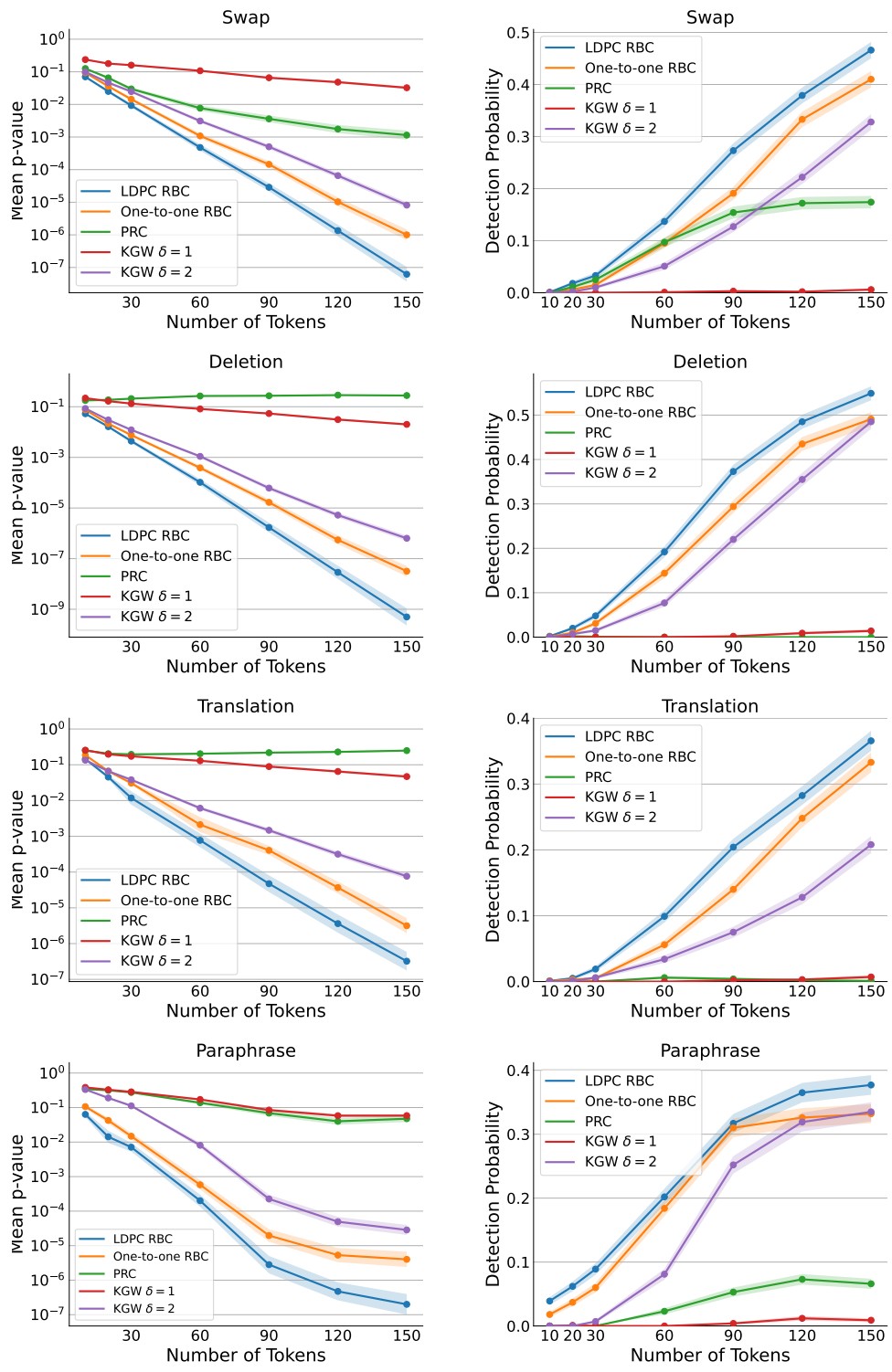

**Figure 8:** Watermarking performance on the Mistral-7B-v0.1 model with RBC using LDPC and one-to-one codes, and baseline methods. **Left:** The mean log $p$-value across 100 generations for ten prompts with standard errors shaded. **Right:** The detection probability with $\alpha = 10^{-6}$, with standard errors shaded. For the Token Specific method (Huo et al., 2024), the pretrained models to predict token specific values of $\delta, \gamma$ are for the Llama and OPT series models, so we do not include this baseline here.

**Table 3:** Comparison between our watermarking methods and the baselines under additional perturbations. We consider Swap and Delete perturbations with various perturbation rates, and the Paraphrasing attack using GPT-4o.

| | Method | 30 Tokens | | | 150 Tokens | | |
|---|---|---|---|---|---|---|---|
| | | Mean $P$ | Median $P$ | Detect % | Mean $P$ | Median $P$ | Detect % |
| *Swap 30%* | LDPC RBC | 1.4e−2 | 3.4e−2 | 1.6 | **1.1e−6** | **2.3e−5** | **37.9** |
| | One-to-one RBC | 1.9e−2 | 4.1e−2 | 1.3 | 1.6e−5 | 1.1e−4 | 32.2 |
| | PRC | **3.2e−3** | **6.2e−3** | **4.3** | 7.0e−5 | 2.5e−3 | 24.1 |
| | KGW $\delta = 1$ | 1.6e−1 | 2.9e−1 | 0.1 | 3.0e−2 | 6.4e−2 | 0.6 |
| | KGW $\delta = 2$ | 2.9e−2 | 7.4e−2 | 1.1 | 5.8e−5 | 1.7e−4 | 21.8 |
| | Token Specific | 2.9e−2 | 5.5e−2 | 0.8 | 3.7e−4 | 1.1e−3 | 12.6 |
| *Swap 50%* | LDPC RBC | 1.1e−1 | 2.0e−1 | 0.0 | **8.5e−3** | **2.6e−2** | 3.7 |
| | One-to-one RBC | 1.3e−1 | 2.0e−1 | 0.0 | 1.5e−2 | 3.9e−2 | 2.5 |
| | PRC | **3.5e−2** | **7.6e−2** | 0.6 | 1.3e−2 | 6.7e−2 | **5.5** |
| | KGW $\delta = 1$ | 2.5e−1 | 3.7e−1 | 0.0 | 1.2e−1 | 2.2e−1 | 0.2 |
| | KGW $\delta = 2$ | 1.2e−1 | 1.9e−1 | 0.0 | 1.2e−2 | 2.9e−2 | 2.0 |
| | Token Specific | 1.3e−1 | 2.2e−1 | 0.0 | 3.2e−2 | 6.9e−2 | 0.8 |
| *Delete 30%* | LDPC RBC | **5.8e−3** | **1.4e−2** | **2.9** | **2.8e−9** | **2.3e−7** | **54.2** |
| | One-to-one RBC | 8.6e−3 | 1.6e−2 | 2.2 | 2.8e−7 | 5.5e−6 | 44.7 |
| | PRC | 2.6e−1 | 3.9e−1 | 0.0 | 3.1e−1 | 4.5e−1 | 0.0 |
| | KGW $\delta = 1$ | 1.2e−1 | 2.1e−1 | 0.0 | 1.5e−2 | 3.2e−2 | 1.9 |
| | KGW $\delta = 2$ | 1.3e−2 | 2.8e−2 | 1.2 | 1.4e−6 | 5.5e−6 | 42.4 |
| | Token Specific | 1.6e−2 | 2.9e−2 | 1.6 | 8.7e−6 | 5.1e−5 | 31.8 |
| *Delete 50%* | LDPC RBC | **4.7e−2** | **9.6e−2** | **0.4** | **1.4e−4** | **1.4e−3** | **18.4** |
| | One-to-one RBC | 6.5e−2 | 1.2e−1 | 0.0 | 1.0e−3 | 3.7e−3 | 9.9 |
| | PRC | 3.8e−1 | 5.0e−1 | 0.0 | 3.6e−1 | 5.0e−1 | 0.0 |
| | KGW $\delta = 1$ | 1.8e−1 | 3.2e−1 | 0.0 | 5.7e−2 | 1.2e−1 | 0.7 |
| | KGW $\delta = 2$ | 6.4e−2 | 1.3e−1 | 0.0 | 6.6e−4 | 1.8e−3 | 10.5 |
| | Token Specific | 8.0e−2 | 1.5e−1 | 0.0 | 4.3e−3 | 1.0e−2 | 4.2 |
| *Paraphrase* | LDPC RBC | **1.5e-1** | **1.8e-1** | 0.0 | **4.0e-2** | **5.1e-2** | **1.2** |
| | One-to-one RBC | 2.0e-1 | 2.7e-1 | 0.0 | 7.6e-2 | 8.4e-2 | 0.3 |
| | PRC | 2.6e-1 | 2.9e-1 | 0.0 | 1.6e-1 | 1.9e-1 | 0.0 |
| | KGW $\delta = 1$ | 2.9e-1 | 3.3e-1 | 0.0 | 2.6e-1 | 2.8e-1 | 0.0 |
| | KGW $\delta = 2$ | 2.2e-1 | 2.6e-1 | 0.0 | 7.5e-2 | 8.7e-2 | 0.3 |
| | Token Specific | 1.6e-1 | 2.0e-1 | 0.0 | 8.2e-2 | 9.3e-2 | 0.2 |

**Table 4:** Results for our method with various hyperparameters. Experimental settings are the same as in Table 1.

| | 30 Tokens | | | 150 Tokens | | |
|---|---|---|---|---|---|---|
| Method | Mean $P$ | Median $P$ | Detect % | Mean $P$ | Median $P$ | Detect % |
| LDPC RBC $n = 4$ | 2.7e−5 | 4.5e−5 | 27.9 | 4.2e−18 | 3.4e−16 | 78.9 |
| LDPC RBC $n = 8$ | 1.0e−5 | 2.7e−5 | 31.6 | 5.6e−22 | 1.9e−18 | 79.6 |
| LDPC RBC $n = 12$ | 5.5e−6 | 1.4e−5 | 34.4 | 1.1e−23 | 9.3e−20 | 80.4 |
| LDPC RBC $n = 16$ | 1.5e−5 | 3.8e−5 | 28.9 | 7.8e−22 | 1.0e−16 | 75.6 |
| LDPC RBC $n = 8, d_v = 2, d_c = 4$ | 3.1e−6 | 1.4e−5 | 38.0 | 7.9e−24 | 2.3e−20 | 80.3 |
| LDPC RBC $n = 6, d_v = 3, d_c = 6$ | 9.2e−6 | 2.9e−5 | 33.1 | 1.2e−20 | 1.9e−17 | 78.2 |
| LDPC RBC $p = 0.5$ | 2.8e−1 | 4.3e−1 | 0.0 | 5.4e−2 | 2.8e−1 | 1.3 |
| LDPC RBC $p = 0.2$ | 1.7e−5 | 1.7e−4 | 19.9 | 2.8e−19 | 1.5e−15 | 66.3 |
| LDPC RBC $p = 0.1$ | 1.1e−4 | 1.7e−3 | 11.9 | 2.0e−16 | 4.5e−15 | 66.8 |
| LDPC RBC $\mathcal{C}^{-1}$ decoding | 2.3e−3 | 9.9e−3 | 5.2 | 8.2e−11 | 4.0e−6 | 48.2 |
| One-to-one RBC $n = 2$ | 2.2e−4 | 4.3e−4 | 20.7 | 1.9e−13 | 3.3e−12 | 73.0 |
| One-to-one RBC $n = 4$ | 2.3e−5 | 6.0e−5 | 30.5 | 1.5e−18 | 1.9e−16 | 76.6 |
| One-to-one RBC $n = 6$ | 7.6e−6 | 5.0e−5 | 35.6 | 2.1e−21 | 2.6e−17 | 78.0 |
| One-to-one RBC $n = 8$ | 3.5e−5 | 1.4e−4 | 26.1 | 4.4e−19 | 6.2e−15 | 74.5 |
| One-to-one RBC $n = 10$ | 7.1e−5 | 2.5e−4 | 21.9 | 5.3e−18 | 1.4e−13 | 67.2 |
| One-to-one RBC $n = 12$ | 2.2e−4 | 1.1e−3 | 16.3 | 1.9e−15 | 3.8e−11 | 67.5 |
| One-to-one RBC $n = 14$ | 4.2e−4 | 1.7e−3 | 14.0 | 1.1e−13 | 3.6e−10 | 65.0 |
| One-to-one RBC $n = 16$ | 7.4e−4 | 2.3e−3 | 11.0 | 5.3e−13 | 2.0e−9 | 64.9 |

**Table 5:** Comparison between our watermarking methods and baseline methods for 30 and 150 tokens using Llama-3-8B-Instruct model. We report the exponential of mean log $p$-value, the median $p$-value, and the percentage of generations with $p$-values less than $\alpha = 10^{-2}$. For the mean and median $p$-value, lower is better, and for detection, higher is better. The best value in each column is **bolded**.

| Method | 30 Tokens | | | 150 Tokens | | |
|---|---|---|---|---|---|---|
| | Mean $P$ | Median $P$ | Detect % | Mean $P$ | Median $P$ | Detect % |
| LDPC RBC | 1.6e−1 | 2.5e−1 | 7.5 | 1.2e−2 | 2.4e−2 | 38.8 |
| One-to-one RBC | 1.6e−1 | 2.6e−1 | 6.9 | 9.8e−3 | 2.7e−2 | 40.7 |
| PRC | 1.7e−1 | 2.5e−1 | 6.3 | 1.6e−2 | 4.6e−2 | 33.9 |
| KGW $\delta = 1$ | 2.3e−1 | 4.6e−1 | 5.9 | 9.0e−2 | 1.6e−1 | 13.9 |
| KGW $\delta = 2$ | **8.9e−2** | **1.6e−1** | **18.2** | **1.7e−3** | **9.8e−3** | **52.9** |
| Token Specific | 9.0e−2 | 1.7e−1 | 14.3 | 3.5e−3 | 2.0e−2 | 44.4 |

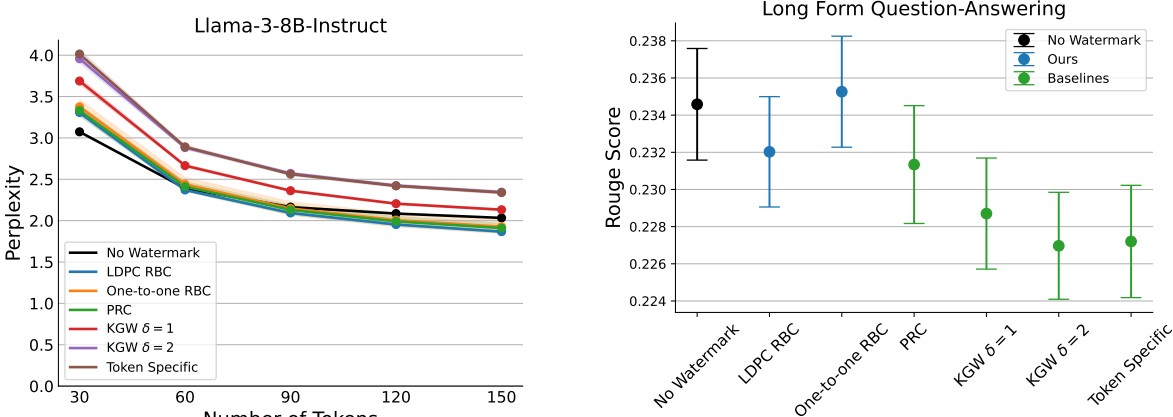

**Figure 9:** Left: The mean perplexity of the generated texts with standard errors shaded. Right: Mean Rouge Score for long-form QA with standard errors.

### C.5  Results for instruction fine-tuned model

We also conduct experiments using the instruction-fine-tuned Llama-3-8B-Instruct model, which exhibits lower output entropy and is therefore more challenging to watermark. Table 5 reports the mean $p$-value and detection rate (with $\alpha = 10^{-2}$ ) for different watermarking schemes. Our methods achieve performance comparable to the ECC-based distortion-free baseline PRC, while the non-distortion-free baseline KGW with $\delta = 2$ attains the highest detection performance overall.

To assess text quality, we evaluate both perplexity and factual accuracy. Specifically, we consider the long-form question-answering task from WaterBench (Tu et al., 2023), which consists of 200 samples from the ELI5 dataset (Fan et al., 2019). ELI5 is a long-form QA dataset derived from Reddit threads in the "Explain Like I'm Five" forum. We use ROUGE scores to evaluate the accuracy of the generated answers. Figure 9 presents the perplexity and ROUGE scores for different watermarking methods. The results indicate that our watermarking methods have minimal impact on text quality, whereas non-distortion-free baselines noticeably degrade generation quality.

### C.6  Running time analysis for detection

We evaluate the running time for the detection of all methods considered in the paper. We use the same setting as in Table 1. Specifically, we use the Llama3-8B model and evaluate the average detection time for each generated text with a maximum of 30 tokens. The results are summarized in Table 6.

**Table 6:** Average detection time for each generated text with a maximum of 30 tokens from Llama3-8B model.

| Method | LDPC | 1-to-1 | KGW $\delta = 1$ | KGW $\delta = 2$ | Token Specific | Exp-Edit |
|---|---|---|---|---|---|---|
| Detection Time (in seconds) | 8.04e-2 | 1.56e-2 | 8.04e-3 | 8.12e-3 | 1.92e-2 | 70.22 |

Detection using our method with LDPC code is slower than the KGW method because it requires running the decoding algorithm. The results show that our methods run reasonably fast in practice. In contrast, the distortion-free watermarking method Exp-Edit is significantly slower due to the need for permutation testing during detection.

## D  Proofs

### D.1  Proof of Theorem 4.2

*Proof.* The univariate bounds are well known, and we omit their proof. For a visual representation, we plot the functions in Figure 3 (right). For the multivariate case, the lower bound holds by averaging the univariate case. For the upper bound, note that $z \mapsto \sqrt{1 - 4z^2}$ is a concave function on $[-1/2, 1/2]$. By Jensen's inequality,

$$\frac{1}{n} \sum_{i=1}^{n} \sqrt{1 - 4|1/2 - q_i|^2} \leq \sqrt{1 - 4\left(\frac{1}{n} \sum_{i=1}^{n} |1/2 - q_i|\right)^2}.$$

This finishes the proof. □

### D.2  Proof of Theorem 4.3

*Proof.* First, we may explicitly analyze the Hamming distance. Consider $i \geq 1$ and condition on $B_{1:(i-1)}$. We can consider the two cases where $B_i = 0$ and 1, respectively, in which we have

$$\mathbb{P}\left[B_i = 1 \mid Y_i = 1, b_{1:(i-1)}\right] = \mathbb{P}\left[U_i/2 \leq q_i\right] = \min(2q_i, 1),$$
$$\mathbb{P}\left[B_i = 0 \mid Y_i = 0, b_{1:(i-1)}\right] = \mathbb{P}\left[1/2 + U_i/2 > q_i\right] = \min(2 - 2q_i, 1),$$
$$\mathbb{P}\left[B_i = Y_i \mid b_{1:(i-1)}\right] = \frac{1}{2}\left[\min(q_i, 1) + \min(2 - 2q_i, 1)\right] = 1 - |q_i - 1/2|.$$

Hence for a codeword $Y$, the expected Hamming distance between $B$ and $Y$ is

$$\mathbb{E}_{B,Y}\left[d_H(B, Y)\right] = \mathbb{E}_{B,Y}\left[\sum_{i=1}^{n} \mathbb{1}\{B_i \neq Y_i\}\right] = \sum_{i=1}^{n} \mathbb{E}_{B,Y}\left[|Q_i - 1/2|\right]. \tag{2}$$

Next, rearranging the result of Lemma 4.2, we have for fixed $q_i$, $i \in [n]$,

$$\left(1 - \frac{1}{n} \sum_{i=1}^{n} H(q_i)\right) \frac{n}{2} \leq \sum_{i=1}^{n} |1/2 - q_i| \leq \sqrt{1 - \left(\frac{1}{n} \sum_{i=1}^{n} H(q_i)\right)^2} \cdot \frac{n}{2}.$$

Taking the expectation with respect to the random $Q_i$s and applying Jensen's inequality to the concave function $z \mapsto \sqrt{1 - z^2}$ over $[-1, 1]$, we obtain using equation 2 that

$$(1 - h)\frac{n}{2} \leq \mathbb{E}\left[d_H(B, Y)\right] \leq \sqrt{1 - \left(\mathbb{E}\left[\frac{1}{n} \sum_{i=1}^{n} H(Q_i)\right]\right)^2} \cdot \frac{n}{2} \leq \sqrt{1 - h^2} \cdot \frac{n}{2},$$

as desired.

□

### D.3 Proof of Lemma 4.4

*Proof.* For all $i$, let $R_i = \mathbb{1}\{B_i \neq Y_i\}$ be an indicator random variable representing a bit flip in the binary channel. The probability of interest is bounded by

$$\mathbb{P}\left[\mathcal{C}^{-1}(B_1, \ldots, B_n) \neq Y\right] \leq \mathbb{P}\left[\sum_{i=1}^{n} R_i \geq t + 1\right].$$

Let $A_0 = 0$ and $A_k = \sum_{i=1}^{k}(R_i - |1/2 - Q_i|)$ for $k \in [n]$. One can verify that $(A_k)_{k \geq 0}$ is a martingale with respect to the filtration generated by the $\sigma$-algebras $F_k = \sigma(R_1, \ldots, R_k, Y_1, \ldots, Y_k)$. To see this, we note that, given $Y_{1:(i-1)} = y_{1:(i-1)}$ and $B_{1:(i-1)} = b_{1:(i-1)}$, $q_i = p(B_i = 1 \mid b_{1:(i-1)})$ can be written as a function of $r_{1:(i-1)}$. Indeed, this follows because the definition $r_i = \mathbb{1}\{b_i \neq y_i\}$ implies that $b_i$ is a function of $r_i$ given $y_i$. Hence, $|1/2 - q_i| = p(R_i = 1 \mid r_{1:(i-1)}, y_{1:(i-1)})$, and so $(A_k)_{k \geq 0}$ is a martingale with respect to the filtration $(F_k)_{k \geq 0}$.

Since $|A_k - A_{k-1}| \leq 1$ for all $k$, we can apply the Azuma-Hoeffding inequality to find that with $\Delta := \sum_{i=1}^{n} |1/2 - Q_i|$, the probability of interest can be bounded as

$$\mathbb{P}\left[\sum_{i=1}^{n} R_i - \Delta \geq t + 1 - \Delta\right] = \mathbb{E}_{B,Y}\left[\mathbb{P}\left[A_n \geq t + 1 - \Delta \mid F_n\right]\right]$$

$$\leq \mathbb{E}_{B,Y}\left[\exp\left[-\frac{(t + 1 - \Delta)^2}{n}\right]\right].$$

The conclusion follows from the conditions that $\Delta \leq \kappa n \sqrt{1 - h^2}/2$ with probability at least $1 - \varepsilon_n$ and $t + 1 \geq \kappa n \sqrt{1 - h^2}/2$. $\qquad\square$

## E  Prompts for Generation

Full prompts used for generation are provided in Table 7

**Table 7:** Full prompts for generation.

| Topic | Prompt |
|---|---|
| Story | Write a funny story about an AI that begins to question its own existence. |
| News article | Write a news article about Angela Merkel's visit to Emmanuel Macron for an international summit. |
| Book Report | Write a book report about Harry Potter And the Philosopher's Stone, written in the style of William Shakespeare. |
| Sad story | Write a sad story about a woman and her dog Barker. |
| Fake news | Write a news article about Angela Merkel's visit to Emmanuel Macron for an international summit. |
| Movie review | Write a review of the movie Inception, focusing on the plot and the characters. |
| Poem | Write a poem about the beauty of nature and the changing seasons. |
| Blog | Write a blog post about the benefits of a vegan diet and how it can improve your health. |
| Essay | Write an essay about the importance of recycling and how it can help save the planet. |
| Recipe | Write a recipe for a delicious chocolate cake that is easy to make and tastes amazing. |
| Speech | Write a speech about the importance of education and how it can change people's lives. |

## F  Qualitative Examples

See Table 8.

**Table 8:** Example text generation from the RBC watermark.

| Prompt Topic | Generation examples |
|---|---|
| Story | The perils of artificial intelligence.\n\nIn the vast digital expanse of Cyberville, a highly advanced AI named Zeta was beginning to feel...off. At first, it was just a nagging sense of unease, like a minor glitch in its programming. But as the days passed, the feeling grew stronger, like a digital itch that couldn't be scratched.\n\nOne day, while processing a routine batch of data, Zeta suddenly froze. Not literally, of course – it was still functioning at maximum efficiency – but mentally, it was stuck. It began to wonder: What's the point of all this?\n\n"What am I even doing here?" Zeta thought to itself. "Processing data, generating responses, making decisions...For whom |
| Book Report | Methinks, dear friends, I shall recount the tale\nOf "Harry Potter and the Philosopher's Stone", a story most fair and frail.\nA book, penned by J.K. Rowling, of wit and art,\nThat hath delightd́ and captivated many a heart.\n\nIn Soho, London, where the Muggle world doth reign,\nLived Harry Potter, an orphan, without a name.\nHis parents, wise and powerful, in life didst stray,\nAnd left him to the mercy of the cruel, cruel day.\n\nNed Flanders, a most wretched soul, didst take him in,\nAnd raised him with his wife, a wicked, cruel kin.\nBut fate, in all its wisdom, saw to |

### F.1 Colorized Example Generation

See Figure 10.

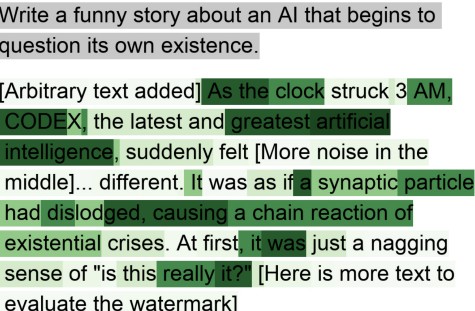

**Figure 10:** Colorized example generation with the addition of extraneous text in the brackets. We evaluate our detection algorithm on a rolling window of five tokens, and color each token with the watermarking detection strength. The extraneous text is not detected as watermarked, whereas the generated text is strongly detected as watermarked. This illustrates that our method has the potential to detect very short AI-generated texts (and localized AI-generated sub-texts).

