# OpenReview forum: "Watermarking Language Models with Error Correcting Codes"
_TMLR — Decision pending for TMLR_

### Review · Reviewer_Gxjg · 2026-03-24

**Summary Of Contributions:**

This paper proposes a watermarking framework RBC that encodes signals through an error correcting code.
They evaluate the RBC on base and instruction fine-tuned models and find that it is robust to edits, deletions, and translations.
and provide an information-theoretic perspective on watermarking, a powerful statistical test for detection and for generating p-values.
Experiments show the method is fast, powerful, and robust, but for long tokens, it is not very good

**Audience:**

Yes

**Audience Explanation:**

This topic is interesting

**Claims And Evidence:**

No

**Claims Explanation:**

I doubt there are some errors, or typos, but they make the theory could not be proved.

1. In Definition 1, either inconsistent of notations or errors.
It says m as message, so m\in\{0,1\}^k, but in the second paragraph, it says $ m\in\{0,1\}^n$.
If $m\in\{0,1\}^k$, then the Hamming ball definition has error, as in d(m,c)<t, m and c has different dimensions, and can not be calculated here.
When define the decoding, is says $d_H (w,c) \le t$, but there is no w, so the definition is invalid.

2. In first 3 paragraphs in Section 2.3, the ambiguous notation Y, either the codeword or one variable bit.

B is a variable, but in Figure 1, B is a vector. So how the get B from Y in Figure 1? B and Y have different dimensions?

3.  Why P[Y=B]=1-2|1/2-q|?
When q\in\{0,1\}, P[Y=B]=1-2|1/2-q|=0, NOT as in the paper P[Y=B]=1/2

4. For simple watermark, it says B and Y are independent, how to generate B?
Why d_H\left(B,Y\right)\approx n/2 if uncorrelated and d_H\left(B,Y\right)<n/2 if correlated?

5. Algorithm 3, what does the output X mean? In this algorithm, only Step 6 update the X by w_{out} bits, along with the original w_{in} bits. For my understanding, for every message, it encode k bits into n bits. But for the output X, it include one “k bits” and several “n bits”, so I’m confused.

6. For Algorithm 4, Step 3 and Step 6, why as that?

7. For Section 4, since I have many questions for preliminaries and algorithms, so I can not check the correctness in this section.
What is the intuitive and straightforward theoretical conclusion?

8. For experiments, it use” e exponential of mean log p-value,”, why not just the mean p-value, that one is not as easy to understand.

**Requested Changes:**

As above

---

> ### Author Response · Authors · 2026-06-11
> **Author Response**
>
> Thank you very much for your feedback. We provide a point-by-point response to your comments below.
>
> > **Comment 1:** In Definition 1 ... so the definition is invalid.
>
> We apologize for the notation inconsistency in Definition 1. In the second paragraph, the notation $m \in \\{0,1\\}^{n}$ was intended to denote an arbitrary binary vector in $\\{0,1\\}^{n}$, rather than a message in the message space $\\{0,1\\}^k$. In addition, in the decoding definition, the symbol $w$ was a typo and should have been the same binary vector.  In the revision, we use a new symbol $v$ for an arbitrary binary vector in $\\{0,1\\}^{n}$, and explicitly introduced the range of the code $\mathcal{R}(\mathcal{C})$. The revised definition is given below.
>
> **Definition 1.1** For positive integers $k\le n$, an error correcting code (ECC)  is an injective map $\mathcal{C}:\\{0,1\\}^k \to \\{0,1\\}^n$. The message space is $\\{0,1\\}^k$. Applying $\mathcal{C}$ to a message $m$ is known as encoding, and $\mathcal{C}(m)$ is known as a codeword. The rate of the code is $k/n$.
>
> Let $\mathcal{R}(\mathcal{C})$ denote the range of the map $\mathcal{C}$. The error correcting distance of $\mathcal{C}$ is the greatest $t>0$ such that for all binary vector $v\in\\{0,1\\}^n$, there exists at most one codeword $c\in \mathcal{R}(\mathcal{C})$ in the Hamming ball of radius of $t$ around $v$, i.e., $d_H(v,c)\le t$. Such a code $\mathcal{C}$ is known as a $[n,k,2t+1]$-code.
>
> Given a $[n,k,2t+1]$-error correcting code $\mathcal{C}$, we may define a decoding map  $\mathcal{C}^{-1}$. For a given $v \in\\{0,1\\}^n$, the decoding $\mathcal{C}^{-1}(v)$ is the preimage of the unique codeword $c\in \mathcal{R}(\mathcal{C})$ such that $d_H(v,c)\le t$, if such a $c$ exists; otherwise, the decoding is defined arbitrarily.
> By definition, at most one such $c$ exists.
>
>
> > **Comment 2:** In the first three paragraphs in Section 2.3 ... have different dimensions?
>
> Thank you for pointing this out. The notation in Section 2.3 was overloaded and may cause confusion. In the algorithmic description, we use the binary language model to generate a bit sequence $B=(B_1,\dots,B_n)\in\\{0,1\\}^n$. The codeword $Y\in \\{0,1\\}^{n}$ is embedded in the sampling scheme. Each codeword bit $Y_i$ is embedded into the sampling process of the corresponding generated bit $B_i$ through the correlated binary sampling channel. Since the sampling mechanism operates bit-by-bit, Section 2.3 first introduces the single-bit case for clarity. In the revision, we replaced the Bernoulli random variables by $B_i$ and $Y_i$ in this section to make clear that the full vectors are generated bit-by-bit.
>
> > **Comment 3:** Why $P[Y=B]=1-2|1/2-q|$? When $q\in(0,1)$, $P[Y=B]=1-2|1/2-q|=0$, NOT as in the paper $P[Y=B]=1/2$.
>
> Thank you for catching this typo.
>
> The correct expression should be $P(Y=B)=1-\left| 1/2-q\right|$. In particular, when $q\in\\{0,1\\}$, the bit $B$ is deterministic and independent of $Y\sim \mathrm{Bernoulli}(1/2)$, which gives $P(Y=B)=\frac{1}{2}$.
>
> The expression can be derived by the following:
> $$
> P(Y = B) =P(B=1\mid Y=1)P(Y=1)+P(B=0\mid Y=0)P(Y=0)
> $$
>
> $$
> = \min\\{2q, 1\\} \times\frac{1}{2} + \min\\{2 - 2q, 1\\} \times\frac{1}{2} = 1-\left|\frac{1}{2}-q\right|
> $$
>
> > **Comment 4:** For simple watermark, it says $B$ and $Y$ are independent, how to generate $B$? Why $d_H\left(B,Y\right)\approx n/2$ if uncorrelated and $d_H\left(B,Y\right)<n/2$ if correlated?
>
> To clarify, in Section 3.1 we do not assume that $B$ and $Y$ are always independent. Rather, the distinction is between the unwatermarked and watermarked settings.
>
> For an unwatermarked bit sequence $B\in\\{0,1\\}^n$, each bit $B_i$ is sampled from the binary language model and is independent of the codeword bit $Y_i\sim \mathrm{Bernoulli}(1/2)$. Then we have
>
> $$
> P(B_i=Y_i)=P(B_i=0)P(Y_i=0)+P(B_i=1)P(Y_i=1)=1/2.
> $$
>
> Therefore,  the Hamming distance (number of mismatches) $d_H(B,Y)$ follows a $\mathrm{Bernoulli}(n,1/2)$ distribution. The notation $d_{H}(B,Y) \approx n/2$ is intended to emphasize that the value is centered around its expectation $n/2$.
>
> In contrast, for watermarked text, Section 2.3 shows that each generated bit is positively correlated with the corresponding codeword bit, with
>
> $$
> P(B_i=Y_i)=1-\left|1/2-q_i\right|\ge 1/2,
> $$
>
> where $q_i$ is the binary language model probability. Therefore, under watermarking, the expected number of mismatches is smaller than in the unwatermarked case, i.e., $\mathbb{E}[d_H(B,Y)]<n/2$, unless the binary language model is deterministic. Theorem 4.3 gives a more precise argument by providing lower and upper bounds for  $\mathbb{E}[d_H(B,Y)]$. Please refer to our response to Comment 7.

---

> ### Author Response · Authors · 2026-06-11
> **Author Response**
>
> > **Comment 5:** Algorithm 3, what does the output $X$ mean? ... so I’m confused.
>
> $X = (X_1, X_2, \dots, X_N)$ is a sequence of tokens. As described in section 2.2, each token is represented by a bit string with length $\ell$, where $\ell=\lceil\log_2\vert \mathcal{V}\vert \rceil$, $\vert\mathcal{V}\vert$ is the vocabulary size.
>
> For every iteration within the while loop in Algorithm 3, we generate $w_{out}$ tokens. In line 4, we use the previous $w_{in}$ tokens to construct a binary message $M$ of length $k$. In line 5, we apply the error correcting code $\mathcal{C}$ to generate the codeword $Y$. In line 6, we generate $w_{out}$ tokens using Algorithm 4. Specifically, we sample $n$ bits using the  Correlated Binary Sampling Channel described in section 2.3, and generate the rest of the bits in the $w_{out}$ tokens using the binary language model directly. The bit sequence is then mapped back to the token space as $w_{out}$ tokens.
>
> So line 6 in Algorithm 3 is used to generate $w_{out}$ tokens, not updating $w_{out}$ bits. Generation of the entire token sequence requires repeats of this procedure. Each iteration uses previous tokens to form a $k$-bit message, and then generates new $w_{\mathrm{out}}$ tokens using the binary language model with the encoded $n$-bit codeword.
>
>
> > **Comment 6:** For Algorithm 4, Step 3 and Step 6, why as that?
>
> In Step 3, we generate the first $n$ bits using the Correlated Binary Sampling Channel (CBSC) introduced in Section 2.3. This step is the key watermarking step: it ensures that each generated bit $B_j$ follows the same Bernoulli distribution defined by the binary language model, and is positively correlated with the codeword bit $Y_j$. Therefore, these $n$ bits carry the watermark signal and can be used for detection.
>
> Step 6 is used for the remaining bits because in general $n$ may not be equal to $w_{\mathrm{out}}\times \ell$. After embedding the watermark into the first $n$ bits, the rest of the bits needed to complete the $w_{\mathrm{out}}$ output tokens are sampled directly from the binary language model. Step 6 is simply the standard Bernoulli sampling procedure.
>
>
> > **Comment 7:** For Section 4 ... What is the intuitive and straightforward theoretical conclusion?
>
> Thank you for this question. The main intuitive conclusion of Section 4 is that the watermarking is easier when the entropy of the language model is high.
>
> As clarified in our response to Comment 3, for a binary bit $B_i\sim \mathrm{Bernoulli}(q)$ generated through the correlated binary sampling channel with codeword bit $Y_i$, the matching probability $P(B_i=Y_i)=1-\left|1/2-q\right|$ is maximized when $q=1/2$, i.e., when the entropy of $B_i$ is largest.
>
> Therefore, when the binary language model has higher entropy, the generated bit sequence $B$ is more likely to align with the codeword $Y$, making the watermark easier to embed and recover.
>
> More formally, Theorem 4.3 shows that the expected mismatch rate $\mathbb{E}[d_H(Y,B) / n]$ is upper and lower bounded by monotone decreasing functions of the average entropy $h$. Thus, larger entropy directly implies fewer expected mismatches.
>
> Since successful ECC decoding only requires the mismatch count to stay within the  error correcting distance, smaller Hamming distance leads to higher decoding success probability. This is formalized in Theorem 4.4, which provides an explicit lower bound on the exact block decoding probability.
>
>
> > **Comment 8:** For experiments ... that one is not as easy to understand.
>
> For watermarked text, most p-values are very close to $0$. In this regime, the arithmetic mean can be dominated by a small number of relatively large p-values and may fail to reflect the strong detection signal across most samples.
>
> To better summarize the overall signal strength, we instead report the exponential of the mean log p-value, which is equivalent to the geometric mean. This summary is more appropriate for highly skewed p-value distributions and places greater emphasis on small p-values.
>
> From a detection perspective, extremely small p-values are particularly important because they correspond to strong evidence against the null hypothesis while maintaining a low false positive rate.

---

### Review · Reviewer_eoB1 · 2026-04-08

**Summary Of Contributions:**

This paper is concerned with watermarking text output by large language models. The general problem is that just by inspecting the text, one should be able to tell that it was produced with the watermark. Moreover, the output of the text should not change in any important way from the output of the model without watermarking. And, the watermark should be relatively robust to small changes in the text (e.g. deleting or rearranging simple phrases).

The paper makes use of error-correcting codes. There are many well-studied error correcting codes with nice properties. Given such a code, the authors give a generic recipe for producing a watermarking system, which they call RBC. Their particular experiments instantiate their watermarks using the low-density parity check (LDPC) code.

Error-correcting codes typically work on sequences of bits. Language models output sequences of tokens. RBC requires some injective map from tokens to bit strings. It takes a sliding window of recent tokens, converts to bits, XORs them with some random bits, and then applies an error-correcting code to the resulting message. They then use a technique they call the correlated binary sampling channel. Given a bit from the codeword Y, and an auxiliary uniform r.v., they sample a bit whose marginal distribution will be the correct Bernoulli distribution (as output by the un-watermarked language model), yet is correlated with the bit from Y. This property alone allows watermarking to be detected (codewords and sampled bits will be uncorrelated in unwatermarked text); the point of the error-correcting code is to make this robust to changes in the text. The authors have some theoretical results connecting the entropy of the underlying language model to the error rate.

They then move to experiments. They test against a variety of baselines from the literature, and against a version of their model that uses a trivial code which doesn't actually correct errors. These experiments involve relatively short generations (there are some example outputs shown in the appendix, which are interesting and which the main body of the paper might benefit from). They find, in general, that their method performs well, and slightly better when a good error-correcting code is used. Performance degrades under changes to the text (e.g. paraphrase, deletion of tokens, translation back and forth, etc.) but degrades least when their full method is used. There are further experiments with multiple language models in the appendix, and the authors suggest further work could involve testing different generation settings as well as a wider range of error-correcting codes.

**Additional Comments:**

I apologize to the authors for the delayed review.

**Audience:**

Yes

**Audience Explanation:**

Watermarking is a topic of general interest and TMLR readers would certainly be interested.

**Broader Impact Concerns:**

There are no broader impact concerns. If research building on this paper does eventually have a broader impact, it will probably be positive (there are few foreseeable harms from accurate detection of LLM text).

**Claims And Evidence:**

Yes

**Claims Explanation:**

The method is clearly described, the experiments while relatively small are sufficiently convincing, and the theoretical results are reasonable.

**Requested Changes:**

I have no substantive requested changes. I felt sections 2.2 and 2.3 were a little hard to understand, and could be rewritten to be much clearer and more consistent in introducing and using notation (these sections are the most important part of the paper at all). This would not be required for acceptance, however.

---

> ### Author Response · Authors · 2026-06-11
> **Author Response**
>
> Thank you very much for your positive feedback. We revised Section 2 accordingly in the revision.
>
> > **Comment 1:** I have no substantive requested changes. I felt sections 2.2 and 2.3 were a little hard to understand, and could be rewritten to be much clearer and more consistent in introducing and using notation (these sections are the most important part of the paper at all). This would not be required for acceptance, however.
>
> Thank you for your suggestion. We have revised Section 2 to make the notation consistent and added Figure 2 to better illustrate the relationship between the generated bit $B_i$ and the codeword bit $Y_i$ under the Correlated Binary Sampling Channel (CBSC). The updates are highlighted in red in the revised manuscript.

---

### Review · Reviewer_R8kN · 2026-05-30

**Summary Of Contributions:**

Contribution 1: a robust text watermark using error correcting codes. This is a marriage of the theoretical ideas in Christ and Gunn (2024) together with a sliding-window encoder in the style of the Kirchenbauer watermark.

Contribution 2: a practical implementation of the ECC-based robust watermark, together with empirical evaluation of this approach to watermarking using the Llama-3-8B model, with comparisons to the Kirchenbauer, Kuditipudi, and Christ watermarks.

My understanding of how this fits in to the literature is:

Kirchenbauer watermark: context -> green list -> bias logits toward green tokens

This work: context -> Y = C(M) -> sample next bits correlated with Y (like Christ & Gunn)

This looks like a nice improvement over Kirchenbauer, which has biased next-token marginal statistics, whereas this approach is marginally next-token distortion-free. My understanding is that next-token distortion-freeness is a weaker statistical guarantee than the sequence-level distortion-freeness guarantee of Kuditipudi, and there is no cryptographic guarantee like Christ.

One downside of the proposed watermark compared to Kirchenbauer is that the Kirchenbauer watermark provides direct hyper-parameter knobs to control distortion vs. watermark power, whereas in this work distortion seems to be a function of the entropy of the prefix string (higher entropy => less distortion). It is not clear to me how to reason about or control the distortion induced by this watermark.

**Audience:**

Yes

**Audience Explanation:**

Yes. I really like the idea of using ECC for robust watermarking. Assuming the exposition and empirical work is cleaned up, this is a nice contribution to the watermarking literature.

**Claims And Evidence:**

No

**Claims Explanation:**

The work is not clearly positioned in the literature. While all the relevant work is cited, it is not easy to understand the connections and departures from previous work. I’ve attempted to map out my understanding of these connections in my summary above; I wish these connections were made more clear in the narrative of the paper itself. I’m also uncomfortable that the discussion of Christ and Gunn (2024) in particular, which is noted by the authors as closely related work, is mostly deferred to the appendix.

Because this watermark does not come with strong statistical or cryptographic distortion-free guarantees, an empirical understanding of the quality of watermarked text is important. The current analysis of sample quality seems confined to the right panel of Figure 3, which doesn’t provide sufficient insight. What language model is used to assess generation perplexity in Figure 3? Is this self-ppl? Generative ppl should be evaluated using a much stronger LM. What’s going on with the strange accent marks in the generated story in Table 8? Is this an artifact of the watermarking process?

I have several concerns below related to the discussion and evaluation of the Kuditipudi watermark that I hope can be clarified before publication.

The paper draws a major contrast between the Kuditipudi watermark and the present work in terms of computational expense. This may be true (computational complexity of Levenshtein distance vs. ECC message extraction and matching) but I don’t think the current comparisons get to the heart of the matter. The primary difference driving computational costs right now seems to be use of a parametric binomial test (present work) vs a non-parametric permutation test (Kuditipudi). The choice of statistical test is peripheral to the watermarking methodology; if a faster parametric approximate test is preferred with higher-resolution (albeit approximate) p-values, something like the Kirchenbauer z-test could be applied to test for the Kuditipudi watermark. Likewise, in principle you could run a permutation test to test for the ECC watermark (to be clear, I am not recommending that you do this!).

The Kuditipudi permutation test is computed with only n=100 permutations and is still reported to be too slow to run for all experiments even with this modest number of samples. This doesn’t sanity-check to me; e.g., the browser demo runs this watermark with n=100 in javascript in about 1 second on my laptop here: https://crfm.stanford.edu/2023/07/30/watermarking.html. The paper reports experiments with n=5000.

Getting to the bottom of the speed issues with the Kuditipudi watermark and reporting comparisons to this watermark for all expts would materially improve this paper. Currently none of the robustness experiments report results for this watermark, and it is the most relevant related watermark because it is specifically designed for robustness. I’d suggest that the authors take another look at the Kuditipudi permutation test, and see whether there is some engineering issue causing it to run so slowly (e.g., are you actually sampling text for the null? That should not be necessary).

**Requested Changes:**

See my comments above. I'd like a clearer narrative about how this work related to other watermarks, and a better understanding about the time-complexity of running the Kuditipudi watermark (ideally resolving this complexity, enabling comparisons to that watermark in the remainder of the expts).

---

> ### Author Response · Authors · 2026-06-11
> **Author Response**
>
> Thank you very much for your feedback. We provide a point-by-point response to your comments below.
>
>
> > **Comment 1:** The work is not clearly positioned in the literature ... is mostly deferred to the appendix.
>
> Thank you for your suggestions. In the revision, we have added more discussion to related work in the introduction.
>
>
> > **Comment 2:** Because this watermark does not come with ... Generative ppl should be evaluated using a much stronger LM.
>
>
> In Figure 3 (Figure 4 in the revision), the perplexity is evaluated using the same Llama-3-8B model that generates the text, i.e., it is a self-perplexity metric. We clarified this point in the revision. We also evaluated text quality using factual accuracy and reported results in Figure 9 in the appendix.
>
> We additionally evaluated perplexity using a stronger language model, Llama-3.1-70B, under the same experimental setting as Figure 3. The results are reported below. The results are consistent with our original conclusion.
>
> | Method                       | LDPC | 1-to-1 | KGW $\delta$=1 | KGW $\delta$=2 |
> | ---------------------------- | ---: | -----: | ------: | ------: |
> | Mean Perplexity (150 tokens) | 6.52 |  19.21 |   22.95 |   24.66 |
> | Mean Perplexity (90 tokens)  | 8.55 |  20.54 |   23.44 |   25.26 |
>
>
> > **Comment 3:** What’s going on with the strange accent marks in the generated story in Table 8? Is this an artifact of the watermarking process?
>
> Sorry for the confusion. This was a typo introduced when transferring the generated text into LaTeX. The intended text was “couldn't” and “what's”. We mistakenly inserted a backslash before the apostrophe, which lead to the strange accent marks. We have fixed the text in the revision.
>
> > **Comment 4:** The paper draws a major contrast between the Kuditipudi watermark and the present work in terms of computational expense ... Likewise, in principle you could run a permutation test to test for the ECC watermark (to be clear, I am not recommending that you do this!).
>
> We thank the reviewer for this insightful comment and agree that the statistical test is conceptually distinct from the watermarking mechanism itself. Our intention in discussing computational cost was mainly to reflect the practical implementations used in prior work, rather than to claim that the computational advantage arises fundamentally from the watermarking mechanism alone.
>
> For our method, we use a parametric Binomial Comparison (BC) test, whose p-values empirically align closely with the false positive rate (Figure 6 in the appendix).
>
> For the Kuditipudi watermark, the detection statistic takes the form $\min_j \mathrm{LevenshteinCost}(\text{text}, \xi, j)$. Deriving an accurate asymptotic distribution for this statistic is nontrivial. A simple parametric approximation (e.g., a z-test) may not provide well-calibrated p-values, particularly in the tail with small p-values required for watermark detection.

---

> ### Author Response · Authors · 2026-06-11
> **Author Response**
>
> > **Comment 5:** The Kuditipudi permutation test is computed with only n=100 permutations and is still reported to be too slow to run for all experiments even with this modest number of samples. This doesn’t sanity-check to me; e.g., the browser demo runs this watermark with n=100 in javascript in about 1 second on my laptop here: https://crfm.stanford.edu/2023/07/30/watermarking.html. The paper reports experiments with n=5000.
>
> According to the official repository (https://github.com/jthickstun/watermark), the web demo uses the detector implemented in https://github.com/jthickstun/watermark/blob/main/demo/detect.js. In the demo implementation, the permutation test generates the random sequence $\xi$ only for the vocabulary appearing in the text to be detected. This design appears intended for computing the null distribution on non-watermarked text. In the official Python implementation (https://github.com/jthickstun/watermark/blob/main/watermarking/detection.py), using the number of observed tokens as the effective vocabulary size is only applied under the null hypothesis.
>
> Our implementation generates $\xi$ over the full vocabulary during detection. Concretely, we use the default sequence length of $256$ for $\xi$, and the Llama-3-8B tokenizer has a vocabulary size of approximately $128$K. As a result, each resample in our permutation test generates roughly $256 \times 128\mathrm{K}$ random numbers. In contrast, the web demo generates approximately $256 \times (\text{number of unique tokens in the text})$ random numbers.
>
> We believe this implementation difference largely explains the discrepancy between the running time observed in our experiments and that of the browser demo.
>
>
> > **Comment 6:** Getting to the bottom of the speed issues ... whether there is some engineering issue causing it to run so slowly (e.g., are you actually sampling text for the null? That should not be necessary).
>
> We report the performance of the Kuditipudi watermark (EXP-Edit) in Table 1. We evaluate EXP-Edit using a p-value threshold of $\alpha=10^{-2}$ for detection with 100 resamples, while all other methods are evaluated at the threshold of $\alpha=10^{-6}$ for detection. Even under this more favorable setting, EXP-Edit exhibits lower detection rates. A fair comparison at $\alpha=10^{-6}$ would likely require at least $10^6$ permutations to obtain sufficient p-value resolution, which would be computationally expensive.
>
> We additionally conducted perturbation experiments for EXP-Edit under the same swap-edit setting as Table 2. We again evaluate EXP-Edit using $\alpha=10^{-2}$ with 100 resamples (versus $\alpha=10^{-6}$ for the other methods). The results are shown below. Under this favorable setting, EXP-Edit does not appear to achieve strong detectability performance in our experiments.
>
>
>
> | Method         |  Mean P | Median P | Detect % |
> | -------------- | ------: | -------: | -------: |
> | LDPC RBC       | 5.3e−10 |   1.0e−8 |     59.0 |
> | One-to-one RBC |  4.6e−8 |   5.8e−7 |     51.3 |
> | PRC            |  2.4e−7 |   3.4e−5 |     41.5 |
> | KGW $\delta=1$        |  9.8e−3 |   2.1e−2 |      1.7 |
> | KGW $\delta=2$        |  4.1e−7 |   1.0e−6 |     49.8 |
> | Token Specific |  4.2e−6 |   1.5e−5 |     35.1 |
> | EXP-Edit       |  6.4e−2 |   6.0e−2 |     24.3 |